# Principles and Applications of Resonance Energy Transfer Involving Noble Metallic Nanoparticles

**DOI:** 10.3390/ma16083083

**Published:** 2023-04-13

**Authors:** Zhicong He, Fang Li, Pei Zuo, Hong Tian

**Affiliations:** 1School of Mechanical and Electrical Engineering, Hubei Key Laboratory of Optical Information and Pattern Recognition, Wuhan Institute of Technology, Wuhan 430073, China; 2School of Mechanical and Electrical Engineering, Hubei Polytechnic University, Huangshi 435003, China; 3Hubei Key Laboratory of Intelligent Transportation Technology and Device, Hubei Polytechnic University, Huangshi 435003, China

**Keywords:** noble metallic nanoparticles, resonance energy transfer, localized surface plasmon resonance

## Abstract

Over the past several years, resonance energy transfer involving noble metallic nanoparticles has received considerable attention. The aim of this review is to cover advances in resonance energy transfer, widely exploited in biological structures and dynamics. Due to the presence of surface plasmons, strong surface plasmon resonance absorption and local electric field enhancement are generated near noble metallic nanoparticles, and the resulting energy transfer shows potential applications in microlasers, quantum information storage devices and micro-/nanoprocessing. In this review, we present the basic principle of the characteristics of noble metallic nanoparticles, as well as the representative progress in resonance energy transfer involving noble metallic nanoparticles, such as fluorescence resonance energy transfer, nanometal surface energy transfer, plasmon-induced resonance energy transfer, metal-enhanced fluorescence, surface-enhanced Raman scattering and cascade energy transfer. We end this review with an outlook on the development and applications of the transfer process. This will offer theoretical guidance for further optical methods in distance distribution analysis and microscopic detection.

## 1. Introduction

In the past few decades, nanomaterials science [1,2,3,4,5] has developed rapidly, and it has formed interdisciplinary subjects with physics, biology, medicine and other disciplines, which have attracted extensive attention and research. Resonance energy transfer (RET) [6,7,8], usually defined as electron energy transfer (EET), is an early-developed optical technology that describes a common photophysical process. In this process, donor molecules excited by electrons transfer energy to acceptor molecules through electrodynamic coupling between transition electric dipole moments, which reflects the transfer process between multiple photo-responsive chromophores. The word “resonance” here implies that there is some correspondence between the excited states of the donor and acceptor. The precise and detailed explanation of energy transfer has been improved over time, and it remains the most dominant and widely used concept in nanomaterials and applications of molecular energy transfer [9]. Owing to the occurrence of, change in and disappearance of RET, the luminescent properties of donor–acceptor pairs will significantly change, including fluorescence intensity, lifetime, etc.

Noble metallic nanoparticles (NMNPs) have excellent resistance in corrosive environments and maintain oxidation resistance even at high temperatures. They were first reported as acceptors for energy transfer in 2001 [10]. In addition to their simple preparation process and stable properties, noble metallic nanomaterials (gold, silver, etc.) have particular properties of localized surface plasmon resonance (LSPR) [11,12,13,14], which makes them available when they participate in energy transfer. The relationship between the donor and acceptor is no longer a simple dipole–dipole interaction, but rather a metal–dipole interaction. In addition, on account of the existence of surface plasmon resonance, the resonance process occurs not only at the dipole–metal boundary but also in the entire nanoparticle structure. The surrounding electromagnetic field will change, and with the influence of the band effect, the bright color display of NPs is enhanced, and the absorption coefficient increases, making NPs good scatterers and absorbers in the range of visible light. Additionally, the anisotropy of the shape also makes the absorption of plasmon resonance possible. Currently, there are a variety of application prospects based on the properties of nanoparticles (NPs), such as drug preparation [15,16,17], bio-imaging [18,19,20], medical sensors [21,22,23], photocatalysis therapy [24], optical data storage [25] and bio-probes [26,27,28,29,30].

In energy transfer involving NMNPs, most of the energy donors are inorganic semiconductor quantum dots or organic fluorescent small molecules, while gold and silver NPs can be regarded as both donors and acceptors. The common energy transfer techniques, which can be divided into fluorescence resonance energy transfer (FRET), nanometal surface energy transfer (NSET), plasmon-induced resonance energy transfer (PIRET), metal-enhanced fluorescence (MEF), surface-enhanced Raman scattering (SERS) and cascade energy transfer (CET), have been widely applied and investigated in numerous fields. Among them, FRET describes the physics of energy transfer between two photosensitive molecules, facilitating real-time dynamic research on molecules under a variety of physiological conditions, but it is usually blind to distances above 10 nm, thus hampering the study of phenomena over long distances [31,32]. NSET has good selectivity for representative metal ions, and it shows unique sensitivity when the distance is more than 10 nm [33]. PIRET involves a strong localized electromagnetic field and the ability of plasmonic nanoparticles to concentrate the light field, which makes them efficient energy donors for constructing transfer systems [34]. Due to the resonance between metallic NPs’ surface plasmons and fluorescent molecules, MEF is regarded as a mirrored dipole, with high sensitivity and efficiency [35]. SERS provides a larger surface area for sites of electromagnetic enhancement, with good biocompatibility and easy modification [36], while in the CET system, the roles of the energy donor and acceptor are no longer fixed, and it is extremely crucial for lower costs and lower energy consumption in research [37].

In recent years, the field has witnessed the rapid development of energy transfer involving NMNPs and their practical applications. In this review, firstly, we present the basic characteristics of NMNPs. Next, we examine and compare the progress of typical RET methods in recent years. Finally, we discuss the prospects for future trends in RET applications, as well as propose new paths for exploring the transfer process between materials and light.

## 2. Basic Characteristics of Noble Metallic Nanoparticles

With the in-depth study of nanomaterials, NMNPs such as gold or silver have been gradually shown to have unique properties in the fields of biology, chemical sensing and nonlinear optics. When the size of NMNPs is reduced to the micrometer or even nanometer scale, the structure will experience numerous physical effects, which makes these materials different from bulk materials in optics. The basic characteristics of NMNPs are as follows:

(1) Physical properties: The optical properties, such as scattering and extinction, of metallic particles attributed to surface plasmon resonance (SPR) were theorized by Mie’s significant contribution in 1908. According to Mie’s work [38], the total cross-section for SPR scattering and absorption is the sum of all electromagnetic oscillations, and appropriate boundary conditions were introduced to describe the SPR phenomenon by solving Maxwell equations. Furthermore, it was pointed out that spherical particles exhibit plasmon resonance, which is due to dipole oscillations of free electrons occupying energy states above the Fermi level in the conduction band. Additionally, these NPs demonstrate various colors, which are due to the different sizes, shapes and kinds of NPs, leading to different absorption characteristics. Figure 1 shows a schematic diagram of LSPR excited by spherical gold NPs, for which only one absorption peak exists, while two pronounced peaks are shown for gold nanorods (NRs), as illustrated in Figure 2. This is due to the longitudinal and transverse plasma bands, which are ascribed to the long- and short-axis oscillation of NRs. Numerous reports have linked the spectral properties of NPs to Mie’s theory, which provided support for the development of metallic band structures, and the optical spectral characteristics of NMNPs have continued to be a hot research topic.

As a typical property, fluorescence quenching can be investigated under various conditions, including femtosecond excitation and the stable state, to explore the interaction between fluorescence groups and NPs. In addition, noble metallic nanomaterials also have large specific surface areas and electrochemical properties.

(2) Chemical properties: Covalent bonds can be formed in the reaction between noble metallic NPs and some mercapto ligands [43], which leads to a substitution reaction.

(3) Radiative properties: Noble metallic NPs scatter strongly when they are near the frequency of SPR. There is a competitive relationship between scattering and absorption. The scattering intensity increases rapidly with increasing volume.

(4) Non-radiative properties [44]: The non-radiative relaxation of plasmons arises from electron–electron collisions (inner- or inter-band excited state) or the surface electron lattice, and photons are absorbed by NPs during scattering.

Owing to the special optical properties of noble metallic NPs, the signal can be captured at the level of only one particle by monitoring the change in plasmon resonance scattering in the process of energy transfer [45,46,47]. Furthermore, the ability of NMNPs to absorb light and convert it to heat makes them good candidates as donors or acceptors in the process of energy transfer. Next, this review will concentrate on the representative resonance energy transfer progress involving NMNPs and compare differences from the point of view of principles and applications.

## 3. Recent Progress in Resonance Energy Transfer Involving Noble Metallic Nanoparticles

### 3.1. Fluorescence Resonance Energy Transfer

In 1949, Förster put forward a theory to describe long-range molecular interactions and derived the transfer rate equation related to the molecular distance and spectral properties [48], so fluorescence resonance energy transfer (FRET) is usually called Förster resonance energy transfer [49,50]. Through interactions between two dipoles, non-radiative energy transfer can be generated, as shown in Figure 3. As a quantum mechanical process, FRET occurs between two fluorescent groups, where energy may transfer from a donor (D) to an acceptor (A) molecule, leading to a reduction in the donor’s fluorescence intensity and an increase in the acceptor’s fluorescence intensity, accompanied by a corresponding shortening and prolonging of the fluorescence lifetime, respectively [51,52,53]. As an ideal donor–acceptor pair for a FRET system, the following four conditions should be met: Firstly, the emission spectrum of the donor should clearly overlap with the acceptor’s absorption spectrum. Secondly, the distance between acceptor and donor fluorophores should be controlled in the range of 1–10 nm. Thirdly, the emission dipole moment of the donor and the absorption dipole moment of the acceptor, according to the separation vector, should have a good mutual orientation. Last but not least, a donor with a high quantum yield is essential.

When the distance between molecules is much smaller than the wavelength, FRET is dominant in the subwavelength range. The FRET efficiency is determined by the distance between acceptor and donor molecules, and it is inversely proportional to the sixth power of the distance. It is calculated as follows [55]:(1)$$k_{T}=\frac{1}{\tau_{D}}(\frac{R_{0}^{6}}{\tau_{DA}^{6}})$$
where $\tau_{D}$ represents the lifetime of the donor along with the acceptor, $r_{DA}$ is the distance between the donor and the acceptor, and *R*_0_ is the Förster radius, which is the distance between the donor and acceptor for an efficiency of 50%. The Förster distance *R*_0_ can be calculated with the following equation [56]:(2)$$R_{0}^{6}=\frac{9000(\ln10)\kappa^{2}\varphi_{D}}{128\pi^{5}Nn^{4}}J(\lambda )$$
where $\kappa^{2}$ is the orientation factor, indicating the relative direction of the donor and acceptor in the space of transition dipoles. Generally, the value is 2/3, which is suitable for randomly oriented dipoles in the donor and acceptor. $\varphi_{D}$ is the quantum yield of the donor, *N* is Avogadro’s number, *n* is the medium’s refractive index, and *J(λ)* is the spectral overlap integral of the donor’s emission spectrum and the acceptor’s absorption spectrum. The spectral overlap integral can be calculated according to Equation (3):(3)$$J(\lambda )=\int_{0}^{\infty }F_{D}(\lambda )\varepsilon_{A}(\lambda )\lambda^{4}d\lambda$$
where $F_{D}(\lambda )$ is the normalized donor’s emission spectrum and is dimensionless, and $\varepsilon_{A}(\lambda )$ is the acceptor’s absorption coefficient; $\varepsilon_{A}(\lambda )$ is expressed in units of M^−1^cm^−1^, and λ is in nanometers. The unit of *J(λ)* is M^−1^cm^−1^nm^4^.

At present, metallic NPs are widely used as substrates and studied with inorganic fluorescent materials to generate FRET systems. Among them, the diameter of gold or silver NPs ranges from 1 to 100 nm, which is similar to most biomolecules’ sizes (proteins, nucleic acids, etc.), so these NPs can be used as probes of the interior of biological tissues to detect the properties of biomolecules, which in turn reveals life processes at the molecular scale. However, the main disadvantage of FRET is the dependence on the photophysical properties of the fluorophores, which depend on the probe’s interaction with the environment and the uncertainty in the position and orientation relative to biomolecules [57].

Dvadyusha et al. [58] used the principle of FRET to design and synthesize a new approach for biosensing based on CdSe quantum dots (QDs) and AuNPs, as illustrated in Figure 4. They performed a hybridization of donor CdSe QDs to the functionalized 5′-end of the DNA strand through covalent bonds and acceptor AuNPs to the functionalized 3′-end. When the hybridization of two complementary strands of DNA occurs, FRET is generated because of the closer distance between the acceptor and donor. The number of complementary DNA chains without nanomodification increases by 10 times; some gold-modified DNA complementary chains are replaced, which prevents FRET when the distance between D-A pairs increases, and the fluorescence signal appears again.

Wang et al. [59] proposed a miRNA detection method based on target-assisted FRET signal amplification, which is ultrasensitive and simple compared with the conventional approach. By using Au nanoparticles as a quencher in the targeted circulatory system, they reduced the detection limit to 1.5 fM, which is of great significance for cancer diagnosis and miRNA research.

Dvadyusha et al. [60] first reported a method to inhibit the interaction between anti-biotin streptin-conjugated QDs and AuNP-conjugated biomolecules using FRET principles, as shown in Figure 5a. Using the interaction between streptavidin and biotin as a model system, QDs and AuNPs were coupled to these two biomolecules, respectively. The fluorescence of QDs is quenched by AuNPs when FRET occurs within the accessible distance through specific interactions. Furthermore, the addition of avidin-saturated biotin-AuNPs prevents the interaction between biotin-AuNPs and SA–QDs and separates them. The FRET effect cannot be generated, and the fluorescence of quantum dots reappears. Li et al. [61] developed a typical apo-Gox-modified gold nanoprobe for the high-sensitivity quantitative analysis of glucose and intracellular imaging research, as illustrated in Figure 5b. FRET led to fluorescence quenching between apo-Gox-modified gold NPs and dextran labeled with fluorescein isothiocyanate (FITC). Once glucose was present, it competed with dextran-FITC. Comparing the affinity of apo-GOx to glucose and dextran, the former is better, and the quenched fluorescence of FITC is recovered. The nanoprobe has a high sensitivity to glucose (detection limit as low as 5.0 nM) and fantastic selectivity.

Wang et al. [62] designed a novel FRET switch sensor consisting of thiazolo [3,2-a]pyridine-7-carboxylic acid (TPCA) and β-cyclodextrin-coated AgNPs (β-CD @ AgNPs) for the rapid determination of an organophosphate pesticide (malathion) in water using non-enzymatic methods. Actually, the targeted molecule can block FRET between the chemical fluorescent probe and β-CD @ AgNPs, enabling precise and accurate malathion detection.

Peng et al. [63] constructed a novel nanoprobe combining AuNPs with a fluorophore for dopamine (DA) detection based on nitrogen fixation, which could achieve dual-signal detection with fluorescence and colorimetric methods, as depicted in Figure 6. With this new probe, the colorimetry can detect DA at concentrations from 0 to 300 M, while the lower limit of the fluorescence method is 0.29 M. The mechanism study showed that the interaction between the AuNP surface and DA’s catechol groups was the key factor for detection. The nanometer probe had good sensitivity and selectivity, and it could realize rapid DA detection.

Similarly, a lead ion (Pb^2+^)-detecting sensor with high sensitivity was constructed by Wang et al. [64] on the basis of FRET between upconversion NPs (UCNPs) and gold NPs (AuNPs) as donors and acceptors, respectively. The FRET process was disrupted due to Pb^2+^-induced aptamers forming G-quadruplexes when Pb^2+^ is present in the environment, and the detection limit was as low as 4.1 nM, which demonstrated high selectivity in practical application, as shown in Figure 7.

Sab. et al. [65] invented an oligonucleotide probe labeled with QDs and rhodamine-immobilized gold nanoparticles (AuNPs-Rh). The detection limit of this new sensitive nano-biosensor reached 3 × 10^−8^ M. This detection method is fast, simple and efficient, without excessive washing and separation steps, as depicted in Figure 8.

Furthermore, various methods have been proposed for exploring FRET. The internal physical mechanism remains one of the common issues that have been challenging to solve, which leads to the accessible distance being less than 10 nm and limits its wide application due to the sensitivity property. Traditionally, the distance between the donor and acceptor is less than 1 nm. At present, an effective way to solve this problem is introducing the surface plasmon nanostructure.

Theoretically [66,67,68] and experimentally [69,70,71], it has been already proven that the local surface plasmon (LSP) regulates the spatial and optical control of the noble metal nanostructure, donor and acceptor during FRET, which are the three constituent units of an LSP. On the one hand, an LSP can independently change the fluorescence intensity and spatial distribution in the donor or acceptor alone; on the other hand, it also affects the near-field reaction between the donor and acceptor, which is the process of non-radiative resonance energy transfer.

The structure adopted for FRET, which is controlled by the LSP, can be divided into two types according to the spatial position of noble metal nanostructures and donor–acceptor pair. The first one is a core–shell structure; generally, the noble metal NPs are the core, while the donor and acceptor are distributed in the shell layer. The other is a planar structure, in which a thin film of the noble metal or a film assembled from NPs is used as a layer, and the donor and acceptor are distributed in another layer or multiple layers.

Using a core–shell structure, Lessard-Viger et al. [72] fabricated a silver core surrounded by silicon dioxide (SiO_2_) as a spacer layer and dye bonding layer. The obtained multilayer concentric structure can simultaneously adjust the distance of donor–acceptor pairs from the Ag core. Meanwhile, due to the presence of the Ag core, the Förster efficiency increases by 4 times, and the range increases by nearly 30%, as can be seen in Figure 9.

Wang et al. [73] constructed a Au nanorod@Ag core–shell nano-system coated with a SiO_2_ monolayer. Rhodamine 590 perchlorate (R590) and oxazine 725 perchlorate (Ox725), two dyes dispersed randomly in the SiO_2_ monolayer, were connected by chemical bonds, as illustrated in Figure 10. The FRET process can be selectively turned on or inhibited by regulating the plasmon wavelength: when the plasmon resonance wavelength coincides with the emission of the donor, FRET is inhibited. FRET is reactivated when the wavelength lies between the emission of the donor and acceptor or coincides with the emission of the acceptor.

For a planar structure, using a layer-by-layer assembly technique, a polymer electrolyte is used as a spacer layer, which can effectively adjust to the distance between semiconductor quantum dots and the layers of the noble metal, which not only makes the adjustment of the layer more precise but also eliminates the influence of energy transfer inside the quantum dot layer, which is caused by an uneven size. Lunz et al. [74] first presented the LSP-mediated FRET process using a sandwich structure, which was composed of the acceptor QD–gold NP–donor QD. Figure 11 shows that the donor and acceptor layers are placed on both sides of the noble metallic layer. Through the reasonable adjustment of the spacing between layers, the Förster radius increased from 3.9 nm to 7.9 nm, the FRET efficiency was 8%, and the rate was enhanced by a factor of 80.

Afzalinia et al. [75] fabricated a biosensor based on the FRET technique and the “sandwich” hybridization of oligonucleotides, as illustrated in Figure 12. In the FRET process, AgNPs and a modified La (III)-metal–organic framework (MOF) were regarded as energy acceptor−donor pairs in the fluorescence quenching procedure. Under optimum conditions, the “turn-off” biosensor can detect the miRNA-155 biomarker at levels as low as 0.04 ppb or 5.5 fM.

Zhang et al. [76] confirmed through experiments and verified by theory that, in this sandwich structure, the coupling between the donor and plasmon is dominant, which is demonstrated in Figure 13a. Tuncay et al. [77] placed the donor and acceptor layers on the same side of the metal nanoparticles so that coupling and energy transfer between NMNPs and quantum dots could achieve selective control. Furthermore, they also found that the non-radiative energy transfer enhanced by plasmon coupling was closely related to the location of coupled donor–acceptor pairs, as shown in Figure 13b.

Bayan et al. [78] proposed electron direct transfer (DT) and the associated FRET process based on two-dimensional graphitic carbon nitride (g-C_3_N_4_) nanosheets and Ag−C_3_N_4_/ZnO plasmonic hybrid heterojunctions. The prepared nanorod heterojunctions exhibited a good photoconductance response, which is due to the energy transfer from g-C_3_N_4_ to ZnO through sufficient band arrangement when the electron concentration is excessive, as illustrated in Figure 14.

In all, as a molecular ruler, FRET involving noble metallic NPs has been widely used in applications such as biological analysis, fluorescence imaging and plasmon sensing. However, FRET is not sensitive when the distance is more than 10 nm, and the study of the long-distance range based on noble metallic NPs is still ongoing.

### 3.2. Nanometal Surface Energy Transfer

Similar to fluorescence resonance energy transfer, when NMNPs act as the energy acceptor for resonance energy transfer, the process can be called nanometal surface energy transfer (NSET). NMNPs or nanopores can accept molecular dipoles as isotropically distributed dipole vectors due to the LSPR effect, opening the curtain on the NSET system. Compared with FRET, the best feature of NSET is that it overcomes the distance limitation, and it shows unique sensitivity when the distance is greater than 10 nm.

Chance et al. [79] studied the energy transfer rate of a dipole transition to a metal surface, and Persson et al. [80] further reported that energy transfer in conductive electrons of metals occurs with different degrees of interaction and distance trends. The dipole–surface energy transfer formula can be defined as $k_{SET}=(1/\tau_{D}){\left(d_{0}/d\right)}^{4}$, and the characteristic distance length can be calculated as:(4)$$d_{0}={\left(0.525\frac{c^{3}\varphi_{D}}{\omega^{2}\omega_{f}k_{f}}\right)}^{1/4}$$
where c represents the speed of light, $\varphi_{D}$ is the donor’s quantum efficiency function, ω is the donor’s electronic transition frequency, and $k_{f}$ and $\omega_{f}$ are the metallic wavevector and Fermi frequency, respectively. The interaction between fluorophores and metal surfaces varies with the distance regime. Generally speaking, the quantum efficiency of energy transfer is
(5)$$\Phi_{EnT}={\left(1+{\left(\frac{r}{r_{0}}\right)}^{n}\right)}^{-1}$$

In the dipole–dipole energy transfer case (FRET), *n* = 6 and *r*_0_ = *R*_0_, and in the case of dipole–surface energy transfer, *n* = 4 and *r*_0_ = *d*_0_ (characteristic distance length) [81].

In a study involving NSET published in 2001, Dubertret et al. [10] described a hybrid nano-system consisting of a single-stranded DNA molecule, 1.4 nm diameter AuNPs and a fluorophore, in which the fluorophore is strongly quenched by AuNPs due to a distance-dependent influence. Figure 15 shows the two conformations of the gold-quenched molecular beacon and the dye–oligonucleotide–gold conjugate. With the addition of the target material, the molecular beacon opens, the distance between AuNPs and the fluorophore increases again, and fluorescence reappears. Although it was not directly stated in their published paper that the result was based on NSET, in fact, it was the first time that researchers used NSET technology to perform analytical detection.

Since then, with the rapid development of gene technology, DNA has been linked to noble metallic NPs with fluorescent molecules, which is being used by more and more people. In 2005, Yun et al. [81] proposed the distance-dependent range for NSET and the efficiency of quenching. They employed a type of dipole–surface DNA that transfers energy from molecular dipoles to the nanometallic surface, with a range more than double that of the traditional system, with an actual distance of 22 nm. Additionally, it was also concluded that efficiency is inversely proportional to the fourth power of the distance, as shown in Figure 16a. In a following study, in a molecular beacon analysis of a hammerhead RNA substrate combined with a catalyst [82], they also measured the kinetics and structure of magnesium ion (Mg^2+^)-induced hammerhead ribozyme conformational changes through NSET for the first time; in addition, it was pointed out that when the distance is greater than 10 nm or dye quenching is used, NSET is more practical and useful than FRET. In 2008, Griffin et al. [83] first studied the monitoring of Mg^2+^ with AuNPs based on NSET to detect the transition state of the RNA unfolding reaction. They confirmed that NSET’s time-dependent ability to clearly distinguish structural transitions between folded and unfolded states also confirmed that NSET is able to track transition states at a distance of more than 10 nm between the donor and acceptor, and the distances, which were used to track RNA folding transition states, were more than double those of the traditional dipole–dipole coulombic energy transfer method, as illustrated in Figure 16b. In the following research, Jennings et al. [84] introduced three dsDNA-dyes with different lengths, which were added with 1.5 nm gold NPs; in addition, they selected dyes with two different energies to measure the quenching efficiency at discrete distances, and a strict quartic relationship between the fluorescence lifetime of the donor and the distance was also found and was consistent with Formula (5). At the same time, 1.5 nm gold NPs did not have an infinite planar structure as theorized but had a high curvature, which made any free electrons within the NPs travel via scattering in the direction perpendicular to the surface. These results showed that gold NPs can be regarded as strongly coupled surface localized plasmons that can be approximated as a planar structure, as illustrated in Figure 16c.

In 2009, Griffin et al. [85] reported the relationship between the size, distance and surface energy transfer characteristics of gold NPs. In addition, NSET was highly correlated with the size of metallic particles, as shown in Figure 17. By selecting gold NPs with different diameters, they tuned the Förster distance R_0_ from 8 nm (which is very close to the reachable distance of conventional FRET (6 nm)) to nearly 40 nm. When the size of the particles increased from 5 nm to 70 nm, the quenching efficiency increased by more than three orders of magnitude.

In the next year, 2010, Chen et al. [86] used gold nanoparticles with different diameters (from 5 to 42 nm) in order to change the distance R, which reflected the distance between the fluorophores at the antibody binding site and the NPs’ center at the aptamer binding site. They demonstrated the NSET system, which possessed long-distance and fast detection capabilities and a simple structure, and a “SET nanoruler” on the surface of living cells was defined. Figure 18 presents a drawing of the SET nanoruler.

In 2015, Prajapati et al. [87] introduced 0.5 mM cetyltrimethylammonium bromide (CTAB) as a surfactant, which increased the distance between single silver nanospheres and silicon quantum dots (Si QDs), resulting in the inclusion of a double-layer CTAB aggregate, which was near the nanospheres, realizing the possibility of changing the NSET efficiency. Figure 19a depicts a schematic diagram of NSET between AgNPs and Si QDs. Cho Tung Yip et al. [88] demonstrated that the competitive relationship between non-radiative quenching induced by surface energy transfer and plasma-electromagnetic enhancement effects sensitized solar cells doped with dye, which has a metal–dielectric–semiconductor core–shell–shell nanoparticle structure. The dye molecule monolayer was chemically adsorbed to the outermost titanium dioxide (TiO_2_) shell. The LSPR of the metal core was excited by light and generated a strong plasmonic near-field around the core–shell–shell NPs, and it was expected that the absorption efficiency of the dye molecules would be significantly improved, as depicted in Figure 19b.

Liu et al. [89] demonstrated the preparation of high-fluorescence amino-functionalized carbon dots via water heat treatment of glucosamine with surplus pyrophosphate for the first time. Figure 20 shows that hyaluronate (HA) molecules on the surface of AuNPs effectively prevent AuNPs from aggregating. In the presence of sufficient hyaluronidases (Hase), HA molecules are broken down into low-molecular-weight fragments, resulting in the aggregation of AuNPs and its exit from the SET system.

Simultaneously, David’s [90] group and Vollath’s [91] group explored the influence of both particle shape and size on surface energy, and they concluded that there exists only a minor dependence on the NP size. Related research is also deepening. Singh et al. [92] explored the spectral overlap of LSPR in the energy transfer process between the dye and gold NPs in the wavelength range of 520 nm to 780 nm. When the fluorescence of the donor overlapped with the LSPR bands of gold NPs, the quenching of the excited state of the measured dye showed a significant dependence on the distance between the donor and acceptor dipole moment, which can be seen in Figure 21. Moreover, they also analyzed the R_0_ value of the NSET process from dye molecules to 2 nm gold nanospheres and compared resonance energy transfer (RET), Gersten–Nitzan (GN) and CPS-Kuhn. It was concluded that NSET was the best model to measure the dye’s quenching efficiency.

Mandal et al. [93] also obtained similar results. By using traditional steady-state spectroscopy, they observed the energy transferred from the surface of P123 to AuNPs. Figure 22a shows a schematic illustration of P123 micelles and probe molecules at different positions. Additionally, a good spectral overlap between the fluorescence spectra of micellar dyes and the LSPR band of gold NPs also satisfies the basic conditions for the occurrence of NSET from the donor dye to acceptor NPs, which is shown in Figure 22b.

Sen et al. [94] first reported that the dye’s PL quenching originated from the non-radiative decay channel in the NSET system, which was based on gold NPs, and the structural change in the protein was measured. During this process, the fluorescence quenching of the dye varied from 0.472 to 0.866, and the radiation rate of the dye was not changed by changing the conformation of the protein. They also confirmed that the distance between gold NPs and dyes was closely related to the hydrophilic dynamics of the BSA protein with different conformations.

Due to the rapid establishment of the theoretical NSET system, it has become increasingly popular for a great number of applications, such as analysis and detection. Li et al. [95] constructed an ultrasensitive fluorescent sensor based on a QD/DNA/AuNP system, as illustrated in Figure 23a. When mercury ions (Hg^2+^) were included in the test solutions, DNA hybridization occurred between two probes, leading to the quenching of QDs. Furthermore, the system exhibited good selectivity for Hg^2+^ in the presence of other metal ions. Krishna et al. [96] prepared a battery-operated ultrasensitive and inexpensive gold-nanoparticle-based NSET probe in order to monitor mercury levels in soil, water and fish. Hg^2+^ ions enabled rhodamine B adsorbed on the surface of gold NPs to desorb, which restored the blocking of the NSET process, as shown in Figure 23b. Similar results were obtained by Sarkar et al. [97]. By developing a silver NP (AgNP)-doped poly (vinyl alcohol)-capped 4-nitrophenylanthranilate (PVA-NPA) complex, they realized the linear detection of Hg^2+^ ions in the range of 0 to 1 ppb and also found a detection limit of 100 ppt, as shown in Figure 23c.

Kurdi et al. [98] presented a method to functionalize AuNPs by combining the supramolecular host molecule curcubit [7] Uril (CB [7]) with rhodamine B (RhB). The fluorescence of RhB was quenched and transferred by AuNPs via surface energy. Then, adenosine triphosphate (ATP) was added, forming a dimeric RhB-ATP complex, and the fluorescence intensity increased ~8-fold. It worked in the concentration range of 0.5–10 μM, and the detection limit for ATP detection was 100 nM.

In sum, as a molecular-scale technology, NSET has been widely used in applications such as optical biological imaging, specific ion detection, quantitative analysis, disease diagnosis, photocatalysis and energy storage, among others, and further in-depth research will broaden its application prospects.

### 3.3. Plasmon-Induced Resonance Energy Transfer

Plasmon-induced resonance energy transfer (PIRET) is a process in which surface plasmons transfer energy from the plasma nanostructure to the adjacent semiconductor through dipole–dipole interactions. Similar to FRET, PIRET has wide applications in biomedical imaging, photocatalysis and molecular-scale analysis.

Compared with dye donor molecules in FRET, NMNPs as PIRET plasmon donors have three main advantages: Firstly, the excitation intensity of the noble metallic NP donors is lower than that required by the dye molecule, which is due to the cross-section of the NMNPs being larger than that of the dye molecule (4–5 orders of magnitude larger than traditional dye). Secondly, compared with FRET, PIRET’s energy transfer efficiency decays more slowly with increasing nanomolecular distances. Last but not least, PIRET is able to enhance molecular fluorescence through long-wavelength excitation light (i.e., lower energy), rather than the excitation light of the molecular absorption peak. Figure 24 [99] illustrates a schematic diagram of energy transfer under different conditions in the NP–molecule hybrid state. (a) In the schematic diagram of energy transfer between two fluorescent groups is shown, due to the presence of the Stokes shift, the acceptor’s emission (green) overlaps little with the donor’s absorption (blue), and no FRET occurs. The partial overlap between donor emission (blue) and acceptor absorption (green) allows FRET to occur. (b) In the schematic diagram of energy transfer between fluorescent fluorophores and NPs, there is no significant Stokes shift in the absorption and emission spectra of NPs (red), which is related to LSPR. Because of the spectral overlap between the dye’s fluorescence (blue) and NPs’ absorption, FRET occurs between them. It can be clearly seen that the absorption peak wavelength of NPs is longer than that of the fluorophores, and the PIRET process is dependent on the overlap between the scattering/emission of NPs and the absorption spectrum of fluorophores. (c) In the schematic diagram of energy transfer between fluorophores and NPs, PIRET does not occur when there is no overlap between the scattering of NPs and the absorption spectra of the fluorophore. However, due to the presence of spectral overlap between dye fluorescence and NPs’ absorption, FRET occurs.

The earliest reports about PIRET date back to 2007. Liu et al. [100] successfully coupled the light scattered by a single particle with its absorption by chromophore molecules on the surface. They found that the PIRET process occurred between cytochrome c and a 30 nm gold nanoparticle. Cytochrome c had multiple absorption peaks in the visible range, among which 550 nm overlaps with the scattering peak of gold NPs to a high degree, ensuring the occurrence of PIRET. When NPs were coupled with chromophores, the energy of the NPs transferred to the molecules. This revealed that the overlap between the scattering peak of the NPs and the absorption by the chromophores resulted in the quenching of NPs, and the position of the spectral overlap was consistent with the absorption spectrum of chromophores. They further proved that the spectral defects generated in the PIRET process were not caused by the direct absorption of scattered light by cytochrome c in the form of radiation, but by the non-radiative transfer process through an electrochemical method. In this process, NPs act as energy donors, while chromophores serve as acceptors.

PIRET can also efficiently harvest sunlight due to its advantages of a large absorption coefficient, wide absorption spectrum and good stability. Li et al. [101] proposed a core–shell nanoparticle system of Au @ SiO_2_ @ Cu_2_O, which can be used to experimentally examine PIRET and FRET between Au and Cu_2_O. FRET and PIRET could be determined by measuring whether the coherent light was transferred from the plasmon to the semiconductor (PIRET) or from the semiconductor to the plasmon (FRET). The results demonstrated that PIRET might effectively use energy below the semiconductor’s band edge to obtain visible and near-infrared sunlight, which is helpful to overcome the limitation of the band-edge energetics of a single semiconductor in photochemical and photovoltaic cells and improve the efficiency of solar energy collection.

Although current research cannot clarify the occurrence and mechanism of PIRET, it is generally believed that the process of PIRET is similar to FRET. Namely, the energy exchange between NPs and small chromophore molecules is carried out through the interaction of dipoles. Choi et al. [102] described the development of an innovative PIRET-based molecular imaging system, which was the first demonstration of in vivo PIRET imaging. In addition, they visualized the dynamics of cytochrome c in HepG_2_ cells through ethanol-induced apoptosis. They deliberately designed the absorption spectrum of dye molecules to effectively overlap with the scattering spectra of AuNPs, which can be seen in Figure 25. Through the observation of the resonance quenching spectrum, the target molecules can be quantitatively and dynamically long-term imaged.

Gao et al. [103] prepared AuNPs with a core–shell structure coated with mesoporous silica, and the injection of rhodamine derivatives triggered PIRET, as shown in Figure 26. As the donor of PRET, the scattering spectrum of AuNPs is around 550 nm, while two different rhodamine derivatives can be selected as acceptors to detect cupric ions (Cu^2+^) and Hg^2+^, respectively. When the closed-ring structure of the prepared rhodamine is consistent with the target, a specific reaction occurs with the ring-shaped rhodamine, with strong absorption at 550 nm. Additionally, through theoretical simulations, they also concluded that the quenching of scattered light was due to the increased dielectric constant of the coupled system.

Similar research was conducted by Cushing et al. [104], who achieved the control of PIRET and hot-electron injection processes in metal @ TiO_2_ core–shell NPs, as seen in Figure 27. Transient absorption spectroscopy showed that plasmon-induced charge separation could be controlled by adjusting the physical contact and spectral overlap between the semiconductor and metal. Furthermore, in the constructed sandwich structure composed of Ag @ SiO_2_ @ TiO_2_, the localized surface plasmon resonance band overlapped with the edge of the TiO_2_ absorption band, leading to PIRET, while the hot-electron transfer process was prevented by the SiO_2_ carrier.

You et al. [105] concluded that the spectral dependence and the asymmetric response of PIRET were due to the interference between the plasmon-generated electric field and external electric field, which can distinguish PIRET from hot-electron injection, with the latter following the shape of the plasmon line. At the same time, a model Hamiltonian approach was used by their group to obtain the optical response and steady-state electron injection rate (SSIR) in the hybrid system [106]. Exciting with a continuous wave, they also found PIRET, leading to a strong SSIR that was completely controlled by the far detuning of the incident light frequency and the transition frequency of the sensitizer.

Hsu et al. [107] deduced the rate expression of Förster-type energy transfer by analyzing the resonance energy transfer in the presence of a plasmonic structure. This theory is applicable to the energy transfer of materials with arbitrary spatial dependence, frequency dependence or complex dielectric functions. At the same time, they also pointed out that PIRET does not necessarily require the maximum overlap of the donor’s emission and the acceptor’s absorption but requires the maximum extinction coefficient in different spectral regions.

Meng et al. [108] successfully realized the solar dehydrogenation coupling of ethane and methane by coupling the Au SPR field with the surface polarization of ZnO nanosheets. Mechanism studies showed that Au-PIRET modulated the stoichiometric conversion by regulating the charge carrier energy, as shown in Figure 28a. Meng et al. [109] found that nitrogen-doped plasmonic Au nanoparticles and La_2_Ti_2_O_7_ (Lanthanum titanium oxide, NLTO) enabled the generation of PIRET by inducing charge separation in NLTO with 600 nm solar radiation. The AuNPs not only acted as photosensitizers but also changed the flat-band potential, inhibited charge recombination and improved the charge extraction efficiency. The Au@Pt-NLTO/reduced graphene oxide (rGO) composite can be seen in Figure 28b. Nan et al. [110] fabricated plexciton-sensitized solar cells (SSCs) based on Au @ chlorophyll (Chl), as illustrated in Figure 28c. Coherent PIRET-efficient channels from metal plasma to molecular excitons enhanced the light collection efficiency of SSCs. The maximal short-circuit current increased by 0.66, while the open-circuit voltage V_oc_ increased by 0.37.

Therefore, as a new direction, PIRET continues to play a consistently significant role in the development of spectroscopy, photonics, biosensors and energy storage devices.

### 3.4. Metal-Enhanced Fluorescence

Since Drexhage et al. [111] first reported that the synchronous emission rate of fluorescent substances could be changed by adjusting the local photon density of states in 1968, researchers have been studying the interaction between metals and luminescent substances [112,113,114]. In reality, metal-enhanced fluorescence (MEF) [115,116,117] is regarded as a mirrored dipole, which is due to the resonance between the metallic NPs’ surface plasmons and organic fluorescent molecules. The energy is effectively transferred between them through non-radiative dipole resonance, and the radiation efficiency of this mirror dipole leads to enhanced fluorescence emission. In essence, the fluorescent substance’s intensity is enhanced compared with that of the free state. As the shape and size of these nanoparticles are difficult to control, the process suffers from high production costs [118].

Guzatov et al. [119] systematically analyzed the overall fluorescence quenching or enhancement of fluorescent materials located near silver spherical NPs, mainly the relationship between silver NPs’ diameters and the distances between the fluorescent substance and NPs, as illustrated in Figure 29a. The maximum enhancement could be achieved at 370 nm excitation light when the diameter of the nanoparticle was 50 nm and the distance from the fluorescent substance was 5–7 nm. In order to make full use of the strong electric field enhancement of the metal and the low dielectric loss, researchers began to pay attention to the metal–dielectric composite micro-/nanostructure to study MEF. Among them, core–shell types, such as a metal@SiO_2_ structure, can effectively improve the strength and stability, which is due to the modulation between fluorophores and the nanometal surface, which is the key to achieving maximum fluorescence enhancement. Lu et al. [120] adopted DNA hybridization technology to propose Ag@ SiO_2_ core–shell nanoflares. When the thickness of the SiO_2_ shell was changed, the fluorophore was limited to the surface. The distance could be precisely controlled, and the enhancement factor in a solution was up to 32 times, as shown in Figure 29b.

Similarly, the transfer matrix method was put forward by Sun et al. to simulate the optimal conditions of the largest fluorescence induced on the metal–dielectric core–shell particles’ surface through large-scale simulations [121]. Plasmonic cores (Ag and Au) and dielectric shells (ZnO, SiO_2_ and Al_2_O_3_) were used, and the optimal radius of each wavelength was determined to achieve maximum fluorescence enhancement in both air and water media, which can be seen in Figure 30. When Au acted as the core, the enhancements could be improved two-fold when the shell refractive index was greater than 2.

Zhang et al. [122] demonstrated PH- and glucose-sensitive swell–shrink properties by synthesizing a core–shell structure with silver acting as the core and cross-linked poly(3-acrylamidephenylboronic acid-co-acrylic acid) acting as the shell (Ag @ PAPBA-PAA). Furthermore, the distance between the Ag core and the fluorophore was controllable, with an enhancement of up to about 1.8-fold. Figure 31a shows a schematic illustration of the pH- and glucose-sensitive hybrid system. Tang et al. [123] reported a similar structure, designing a nano-system on the silver nanoparticle with hybrid poly (N-isopropylacrylamide-co-acrylic acid) (PNIPAM-co-PAA) microgels, as illustrated in Figure 31b. Adjusting the interaction distance between the AgNPs and fluorophores changed the shrinkage behavior of the microgels, which was caused by the swelling and shrinkage of hybrid microgels in response to a stimulus.

Aslan et al. [124] synthesized a Ag@SiO_2_ core–shell structure that binds any fluorophore to the SiO_2_ shell. When an organic fluorophore (Rh800) was linked to the shell, the fluorescence signal was enhanced by 20 times (with Rh800) and the particle detectability was increased by 200 times, as illustrated in Figure 32.

On this basis, Asselin et al. [125] developed nanostructures of Ag@SiO_2_@SiO_2_ + X (x was a specific fluorescent dye, such as rhodamine B or eosin) and varied the size of the core, the thickness of the silica shell and the concentration of the fluorophore. They derived the conditions under which the key parameters affected the optimal values through traditional spectroscopy, such as time-resolved fluorometry, UV-VIS spectrometry and transmission electron microscopy.

With the development of the mechanism of MEF, researchers have prepared various fluorescent substrates through different methods. Yang et al. [126] fabricated a surface-enhanced fluorescent substrate with the adjustable size and spacing of silver nanospheres using a combination of nanoscale printing technology and electro-deposition, and the optimal fluorescent enhancement was 17 times. They also put forward that the enhancement effect relied on the size of NPs and was closely connected to the degree of overlap (coupling efficiency) between the characteristic spectra of fluorescent molecules and the LSPR of metal nanostructures.

Fu et al. [127] used the seed growth method to prepare gold nanorods with an average length of 80 nm and an average diameter of 13 nm in an aqueous solution, which contained cetyltrimethylammonium bromide (CTAB), and Cy5 ssDNA molecules connected with gold nanorods, as can be seen in Figure 33. The fluorescence intensity of the Cy5 ssDNA molecule was about 40 times stronger than that of unconnected gold nanorods, and the excitation light was 633 nm. Moreover, it was pointed out that the regulation of the aspect ratio could be measured at the single-molecule level for biological detection.

Zhao et al. [128] studied the spectra of organic fluorescent molecules based on Au nanorods. They found that when the length of Au nanorods increased, the peak was at lower energy and red-shifted linearly. In addition, a new plasmon resonance peak was observed. These results can promote the understanding and application of the MEF effect.

Yun et al. [129] proposed a novel MEF-based biosensor system consisting of fibrous Ag@SiO_2_ polycaprolactone (PCL) substrates. The preparation process included the Stober process, photoreduction and electrospinning, as shown in Figure 34a. The silica layer acted as a spacer between the AgNPs and the fluorescent molecules to optimize the MEF effect. In addition, they further improved the MEF effect by controlling the shape, size and type of metal-based nanostructures distributed on the fibers. Xu et al. [130] fabricated a hybrid nano-system with Au nanoclusters (NCs) conjugated to the surface of Ag@SiO_2_ core–shell. These core–shell-structured NPs achieved MEF, with good water dispersibility, easy conjugation, excellent stability and strong fluorescence emission. When the separation distance was about 10 nm, the fluorescence enhancement could be up to 3.21-fold. Based on the switching characteristics of MEF, it was further applied to the detection of Cu^2+^ and inorganic pyrophosphate (PPi), as illustrated in Figure 34b.

Furthermore, the enhanced fluorescence signal can also be used for smartphone-based surface-plasmon-coupled emission (SPCE) [131,132,133]. As a prism-coupled technology that couples fluorescence to the SPPs of metal films, SPCE can achieve signal collection efficiencies larger than 50% due to the special directivity of emission [134].

Since the early achievement by Lakowicz’s group [135,136], SPCE technology has been applied in the development of several biosensing platforms. In addition to the high p-polarization properties of the transmitting signal, SPCE shows high background and spectral resolution [137]. Rai et al. [138] obtained over 900-fold SPCE enhancement with a plasmonic–dielectric hybrid on an SPCE platform. The same group [139] proposed Soluplus-mediated AgAu nanohybrids with enhancements of >1200-fold. Given the biocompatibility of these synthetic NPs, this approach will be widely used in near-real-time management strategies in numerous areas for the point-of-care diagnosis of diseases [140].

Commonly, research on MEF will be enhanced with existing biosensing methods related to cryosoret technology [134,141], ferroplasmon-on-mirror (FPoM) technology [134,142] and photonic-crystal-coupled emission (PCCE) platforms [134], as is shown in Figure 35. These technologies and related sensing platforms are expected to improve current biosensing models to better understand biophysical chemical processes and related outcomes at advanced micro–nano interfaces.

With the study of plasmon-enhanced spectroscopy, the plasmon resonance characteristics of the metal surface have been widely used to realize the breakthrough of enhanced fluorescence. This kind of surface-enhanced fluorescence with high sensitivity, high efficiency and convenience has become an important research direction in the field of nanophotonics and plasmon sensing.

### 3.5. Surface-Enhanced Raman Scattering

In the 1970s, Fleischmann et al. [144] performed a Raman spectroscopy test on the surface of a rough silver electrode to which pyridine molecules were attached. Thus, for the first time, they obtained a Raman spectrogram of the monolayer of a pyridine molecule. Afterward, Van Duyne [145] and J Alan et al. [146] found that when pyridine molecules were adsorbed on the surface of the rough silver electrode, the measured Raman scattering was about 10^6^ times stronger than the measured Raman scattering signal in a solution, according to a large number of experiments and systematic theoretical calculations. They considered that this was a new kind of surface-enhanced effect caused by the rough surface and defined it as surface-enhanced Raman scattering (SERS). The corresponding spectrum was the SERS spectrum.

Over the years, researchers have conducted a large number of studies in the field of SERS-active substrates. The earliest SERS-active substrates were rough metal electrodes and metal island films [147]. The size and morphology of the nanostructure were difficult to control, and the enhancement effect was limited. Therefore, they were not ideal SERS-active substrates. In 1979, silver and gold nano-sols were first applied to SERS as active substrates. To date, noble metal nanospheres with different particle sizes and nanorods with different length–diameter ratios have been successfully designed by the chemical reduction method [148]. The Raman scattering signal is greatly enhanced by molecules adsorbed on the metal NPs’ surfaces. In contrast to standard Raman scattering, SERS requires metal nanostructures as substrates; silver and gold are commonly used as substrates. Silver has an ideal electromagnetic enhancement effect and excellent plasmon efficiency in the visible-light range, and it tends to produce higher SERS enhancement at a relatively low cost [149], while gold is characterized by easy modification, high stability and good biocompatibility. Moreover, both the advantages and disadvantages of SERS are clear. The main disadvantages derive from its inherent lack of reliability and universality [150].

As new SERS substrates, metal nanocomposite structures have the physical properties of multiple materials and have been widely studied. Anema et al. prepared metal NPs with a core–shell structure coated with a thin silicon dioxide film as an active substrate, which not only retained the structure’s electromagnetic enhancement effect but also eliminated the spectral interference caused by the metal surface being exposed [151].

Samal et al. [152] described a size-controllable Au @ Ag core–shell nanosphere structure using the efficient and simple water synthesis method, as illustrated in Figure 36a. Under near-infrared light excitation, the SERS enhancement effect was significantly improved compared with Ag nanospheres of similar size. Through the co-reduction of ascorbic acid (AA), Yang et al. [153] prepared Ag-Au hollow nanostructures via the electric displacement reaction between Tetrachloro-auric acid (HAuCl_4_) and Ag nanocubes. The SERS enhancement effect and chemical stability were greater than those of individual silver nanocubic components, as shown in Figure 36b.

Chang et al. [154] observed SERS with maximum intensity on Au nano-gratings with a period of 200–400 nm. Figure 37 shows a schematic diagram of the SERS and fluorescence signals of two nanograting structures: (a) non-lift-off and (b) lift-off. The enhancement had a periodicity dependence. The enhancement of these two structures was attributed to the plasmonic effect, and the excitation of SPP led to a larger SERS effect and enhancement.

Qu et al. [155] analyzed the chemical information of red wines with SERS through several sample preparation methods. An innovative method that suppressed adenine’s dominance in the SERS spectrum was presented, and important phytochemicals were extracted from red wines when they were concentrated on in situ prepared AgNP mirrors, as illustrated in Figure 38.

Chang et al. [156] developed a SiO_2_ @ Ag core–shell structure that was modified with gold nanospheres to create isotropic hot spots to form a heterosatellitallic shell–satellite structure, which acted as the SERS probe, and the enhancement factor was as high as 10^6^, which can be seen in Figure 39.

In addition, SERS technology has significant advantages in the field of cell monitoring. Xu et al. [157] presented a synthetic method for the selective DNA modification of nanorods (NRs), followed by the assembly of nanospheres (NPs) to form complex assemblies. Three types of structures, terminal, side and satellite structures, were synthesized with a yield of over 85%. A variety of experimental methods proved the stability and uniformity of the various assemblies, as shown in Figure 40. The fine control of geometrical structure, the smaller inherent size compared with cellular organelles and the presence of gaps between particles in these assemblies lead to SERS in cells. The culture of HeLa cells with unlabeled NP-NR assemblers produced sufficient SERS signals to detect lipids and small metabolites in the mitochondrial structure. It was also the first time that conceptual validation data were provided, which revealed the possibility of the real-time monitoring of the organelle environment in living cells.

Furthermore, Kim et al. [158] reported a new biosensor, which adopted the enhanced Raman spectroscopy of the Ag/Polydiacetylene (PDA) core–shell hybrid system as complementary signals to collect DNA information. Figure 41 shows a schematic diagram of the single-stranded DNA hybridization process with a Ag/PDA core–shell nanoparticle and PDA nano-shell. The PDA’s luminescence changed from green to red under dry conditions without the use of additional dyes, and the photoluminescence efficiency was significantly improved due to the plasmon enhancement effect.

Zhang et al. [159] analyzed the different catalytic activities of thiophenol on silver, gold and copper nanoparticles by studying the SERS spectra of the products reduced by hot electrons. By monitoring the efficiency of the reaction by in situ SERS, it was demonstrated that six-electron reduction could also occur, indicating that the ligands may be considered promoters instead of mandatory elements in the reaction.

In short, SERS plays an important promoting role in life science, chemical materials, optical physics and other related disciplines due to its many advantages.

### 3.6. Cascade Energy Transfer

As for traditional RETs, when the donor chromophore is stimulated, part of the energy obtained is transferred to the energy acceptor in a non-radiative way, thus completing the whole energy transfer process. In the process, the presence of energy donors and acceptors is relatively fixed, and the energy is transferred in a single direction (namely, from donor to acceptor). In fact, there is also another kind of energy transfer system, that is, cascade energy transfer (CET) [160,161,162,163]. In CET, there is more than one D-A pair, as illustrated in Figure 42 [164]. The acceptor chromophore in each D-A pair may be the donor chromophore in the next pair. Therefore, the roles of the energy donor and acceptor are no longer fixed in the new system.

The common different cascade energy transfer mechanisms can be divided into three types [165]: (1) there are two FRET steps between the first and second chromophores and between the second and third chromophores; (2) FRET occurs from one individual donor to two different acceptors; (3) as in the first case, there are still two steps of FRET, but they are between the first and third chromophores, which can be seen in Figure 43.

Cascade energy transfer involves more than one set of FRETs; this multistep energy transfer system has some advantages compared with single-step energy transfer: for example, a higher efficiency in a wide wavelength range, a larger Stokes shift and the easier detection of the final acceptor, though this process may need a longer reaction time.

Belusákova et al. [166] described the multistep FRET process of laser dyes on NPs. They selected six cationic laser dyes whose spectral properties were suitable for resonance conditions, and the possible relaxation process of the six dyes was also discussed, as illustrated in Figure 44.

Tsukamoto et al. [167] investigated the energy transfer of three coumarin @ (OAm)_2_ complexes, in which the cascade occurred between dyes coated on the surface of a nanosheet. The energy transfer rate was also systematically studied, as shown in Figure 45.

The CET process based on the participation of metal NPs is worthy of further exploration for bionic optical synthesis, bioactivity analysis and cell membrane multivalence.

It can be clearly observed that the above resonance energy transfer process is tied to the participation of noble metal NPs, as illustrated in Figure 46.

In sum, the different resonance energy transfer processes involving plasmonic NPs from noble metals with their advantages, disadvantages and applications are shown in Table 1.

## 4. Conclusions and Perspectives

In conclusion, resonance energy transfer involving noble metallic NPs has become the focus of extensive applications in various fields. Here, the research progress based on resonance energy transfer involving noble metallic NPs, including fluorescence resonance energy transfer, nanometal surface energy transfer, plasmon-induced resonance energy transfer, metal-enhanced fluorescence, surface-enhanced Raman scattering and cascade energy transfer, is extensively reviewed, and the advantages, disadvantages and applications are all presented. This paper shows that the process of energy transfer has been not only used for synthesis and functionalization but also preliminarily proven to be superior for realizing analysis and detection to develop new transfer systems, which have important research value and practical significance.

However, although many attempts have been made to apply resonance energy transfer to practice, there are still many unsolved problems with resonance energy transfer involving noble metallic NPs. Future work can be further expanded and deepened from the following aspects:

(1) Under different energy transfer conditions, methods to adjust the physical morphology of the NPs and optimize the distance and coupling system between the NPs and the fluorescent group are frequently discussed. However, the way to obtain optimal transfer efficiency remains an urgent research target.

(2) The mechanism of energy transfer between the local electric field around the NMNPs, the emission field near the fluorescent group and the emission field of the incident light remains to be further explored.

(3) Methods to select the appropriate structure, such as core–shell, flat layer, sandwich and other models, to obtain the optimal efficiency still needs more attention.

(4) A series of energy conversion processes may be triggered in different kinds of energy transfer, such as photon-thermal-electricity [107] and photon-electricity-thermal [108] processes. This provides technical guidance for further exploring the physical mechanism of energy transfer and developing micro-optoelectronic devices.

## Figures and Tables

**Figure 1 materials-16-03083-f001:**
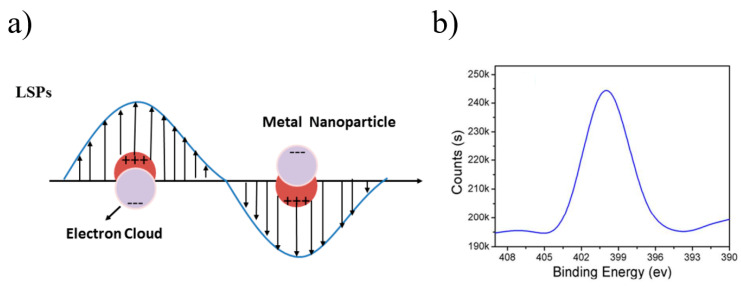
(**a**) Typical localized surface plasmon resonance of gold nanospheres [39]; (**b**) absorption spectrum of gold nanospheres [40].

**Figure 2 materials-16-03083-f002:**
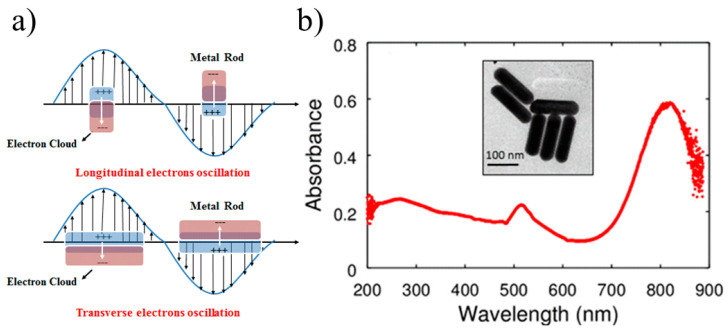
(**a**) Two typical forms of localized surface plasmon resonance (longitudinal and transverse electron oscillations) of gold NRs. Adapted from Ref. [41]; (**b**) absorption spectra of gold NRs [42].

**Figure 3 materials-16-03083-f003:**
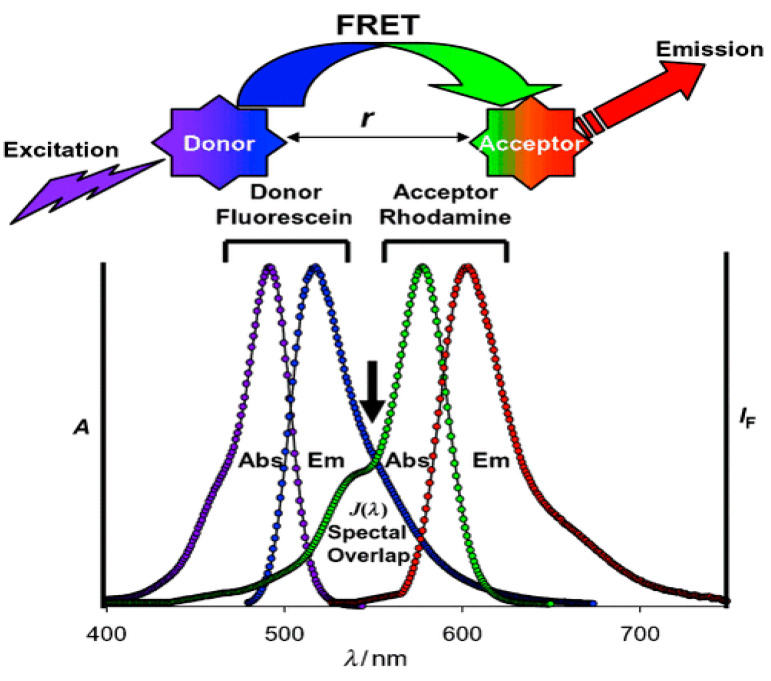
Schematic representation of FRET system [54].

**Figure 4 materials-16-03083-f004:**
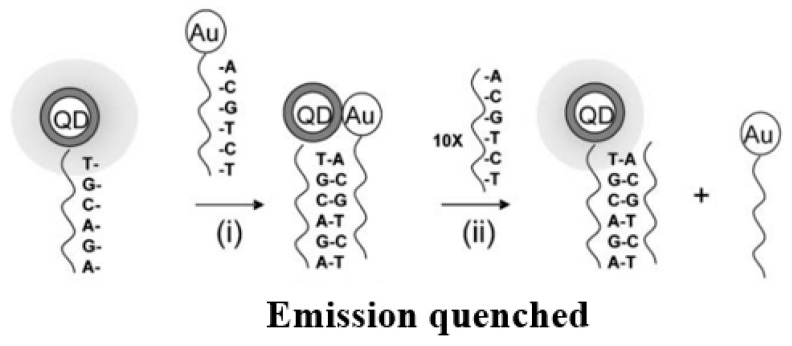
Schematic diagram of QD–DNA biosensor. (i) Addition of 1 equivalent of the Au-DNA to the QD-DNA to yield the hybrid, (ii) Addition of 10 equivalents of the unlabelled complementary oligonucleotide to the hybrid [58].

**Figure 5 materials-16-03083-f005:**
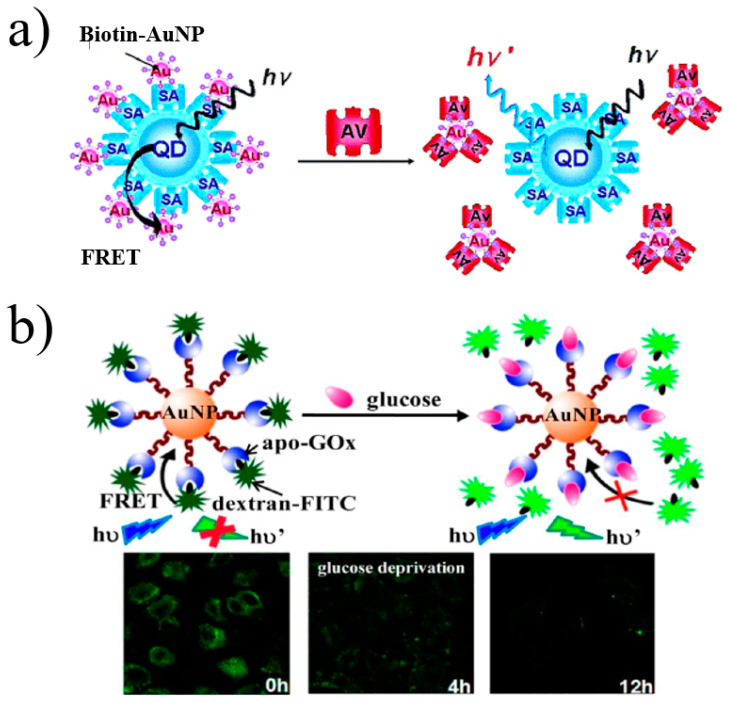
(**a**) Schematic of a method to inhibit the interaction between anti-biotin streptin-conjugated QDs and AuNP-conjugated biomolecules [60]. (**b**) A typical AuNP-apo-Gox-dextran-FITC nanoprobe [61].

**Figure 6 materials-16-03083-f006:**
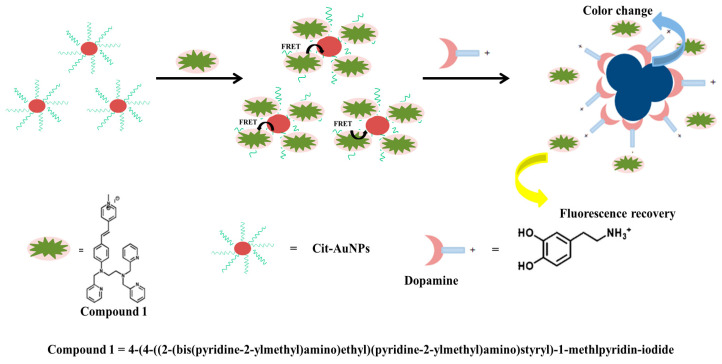
Schematic illustration of a novel nanoprobe containing AuNP fluorophore for detection of DA. Adapted from Ref. [63].

**Figure 7 materials-16-03083-f007:**
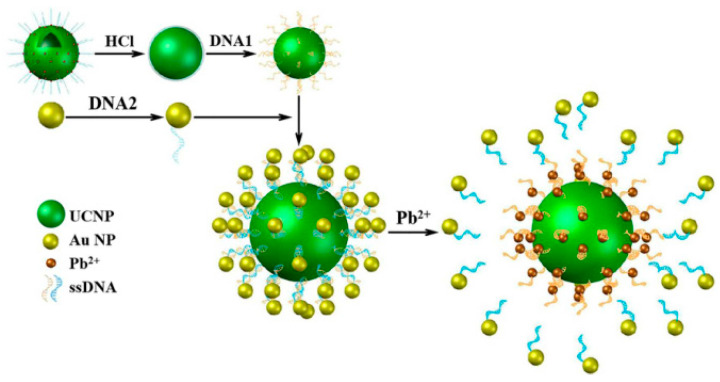
Schematic illustration of a Pb^2+^-detecting sensor with high sensitivity [64].

**Figure 8 materials-16-03083-f008:**
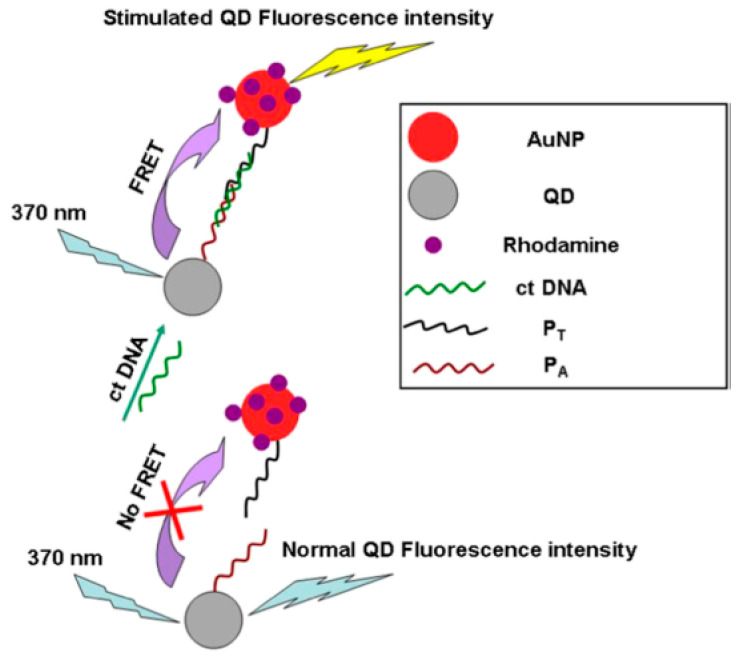
Schematic of a newly developed FRET-based sensor [65].

**Figure 9 materials-16-03083-f009:**
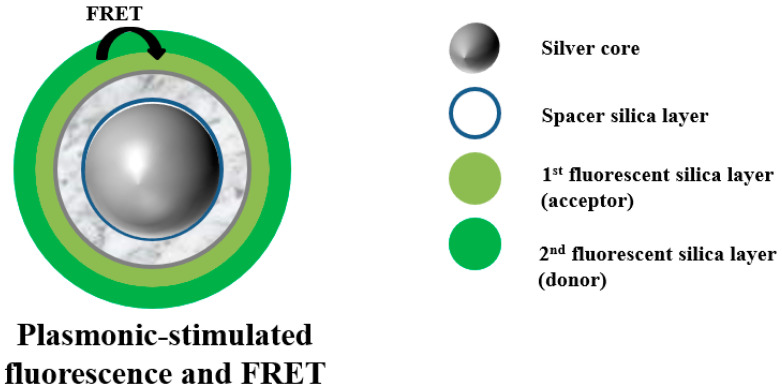
Schematic representation of fluorescence and FRET excited by plasmon. Adapted from Ref. [72].

**Figure 10 materials-16-03083-f010:**
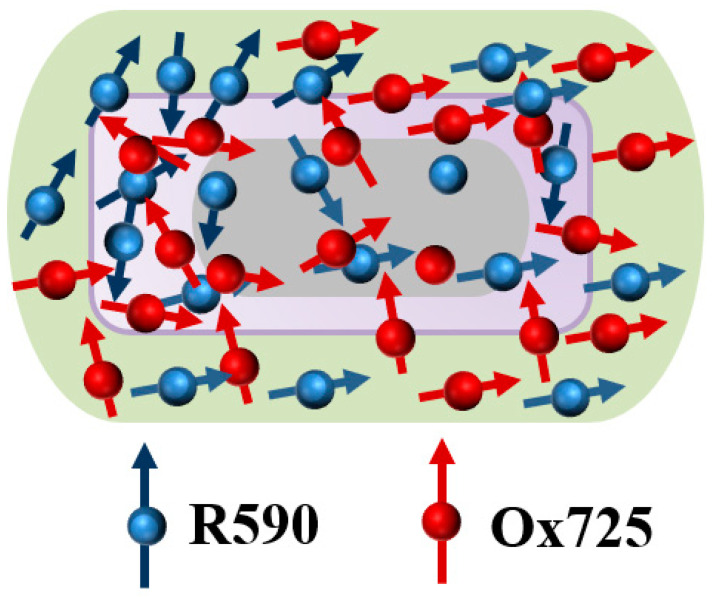
Schematic showing the Au nanorod@Ag core–shell nano-system coated with SiO_2_ monolayer. Adapted from Ref. [73].

**Figure 11 materials-16-03083-f011:**
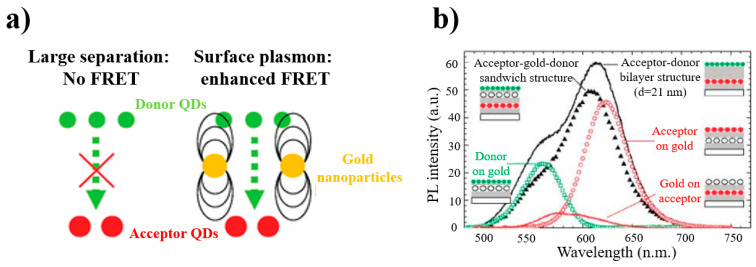
(**a**) Schematic diagram of energy transfer between CdTe regulated by surface plasmons of AuNPs. (**b**) Photoluminescence spectrum of this sandwich structure (black triangles) with different gold and QD configurations. [74].

**Figure 12 materials-16-03083-f012:**
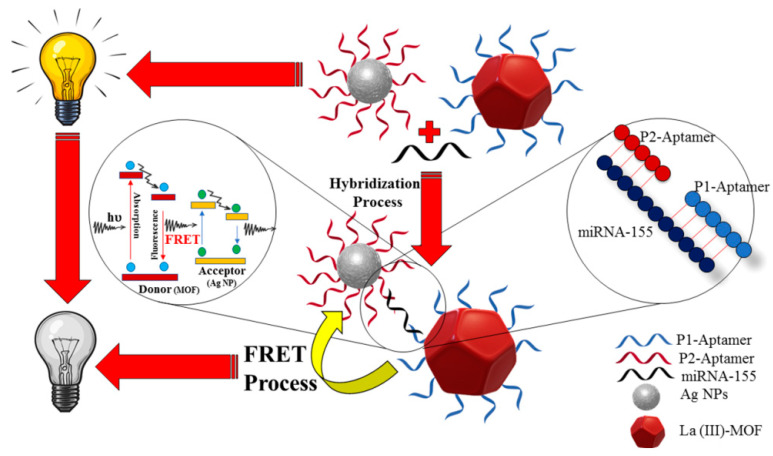
Schematic diagram of a biosensor based on detection of miRNA-155 [75].

**Figure 13 materials-16-03083-f013:**
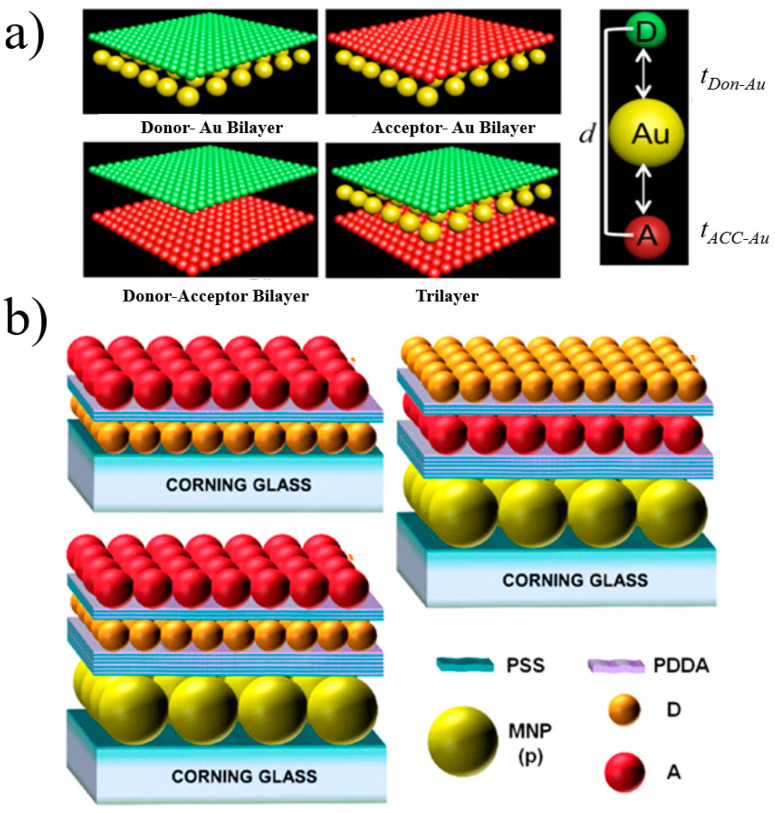
(**a**) Schematics of typical sandwich structures [76]. (**b**) Layered architectures of different FRET systems [77].

**Figure 14 materials-16-03083-f014:**
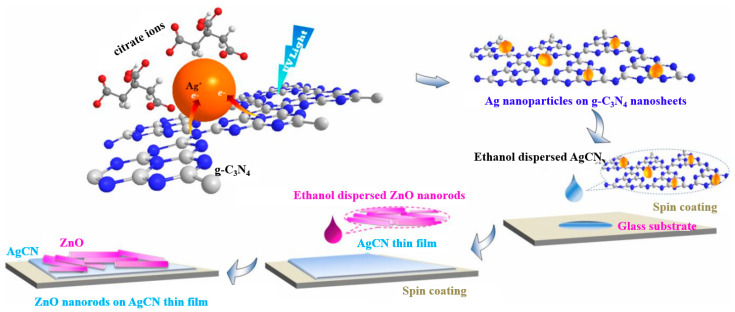
Schematic representation of the synthesis procedure of plasmonic hybrid heterojunctions [78].

**Figure 15 materials-16-03083-f015:**
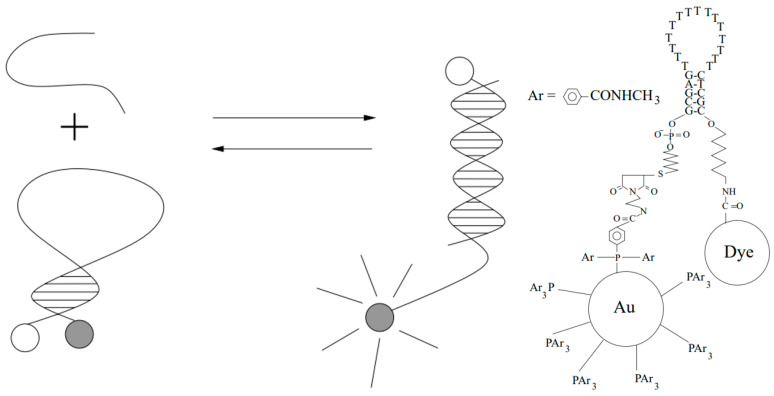
Two conformations of the gold-quenched molecular beacon and the dye–oligonucleotide–gold conjugate [10].

**Figure 16 materials-16-03083-f016:**
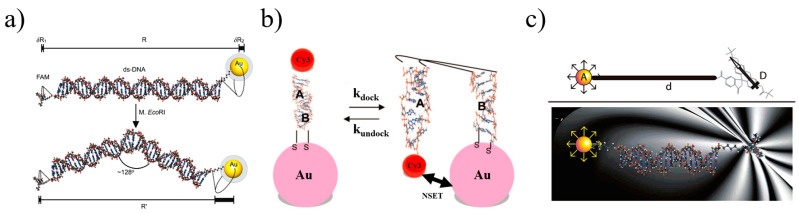
(**a**) Schematic drawing of the NSET system, which consists of gold NPs and fluorophores appended to dsDNA [81]. (**b**) Hairpin ribozyme of the docking and undocking transitions [83]. (**c**) Schematic diagram of a donor-dye–nanometal-acceptor pair separated by dsDNA [84].

**Figure 17 materials-16-03083-f017:**
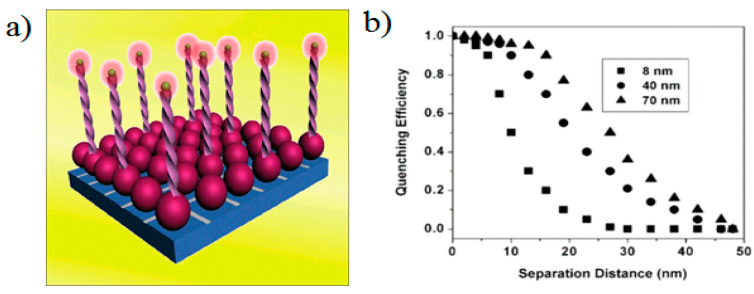
(**a**) Schematic drawing of relationship between NSET’s scale and distance. (**b**) Effect of distance between gold nanoparticle and Cy3 dye on quenching efficiency [85].

**Figure 18 materials-16-03083-f018:**
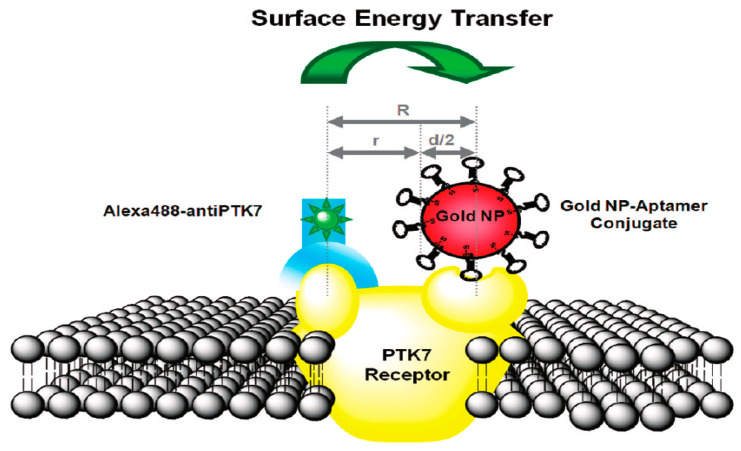
Drawing of a SET nanoruler [86].

**Figure 19 materials-16-03083-f019:**
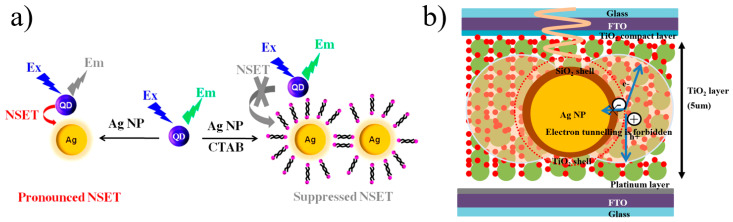
(**a**) Schematic diagram of NSET between AgNPs and Si QDs [87]. (**b**) Schematic drawing of the core–shell–shell nanoparticle structure. Adapted from Ref. [88].

**Figure 20 materials-16-03083-f020:**
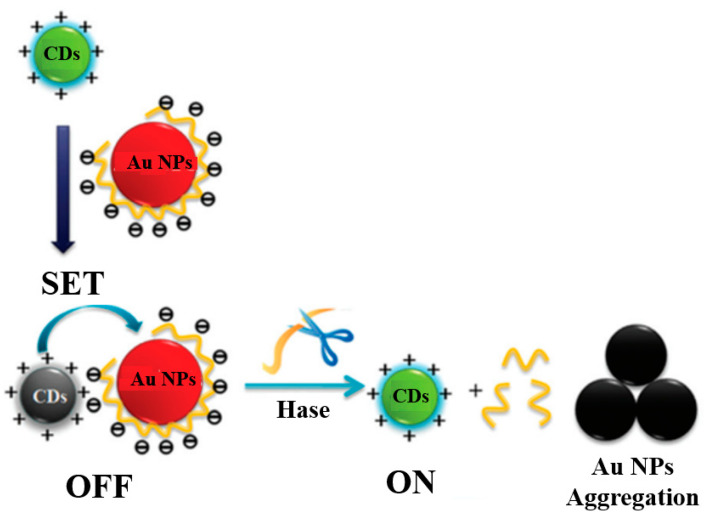
Schematic illustration of a novel SET biosensor system [89].

**Figure 21 materials-16-03083-f021:**
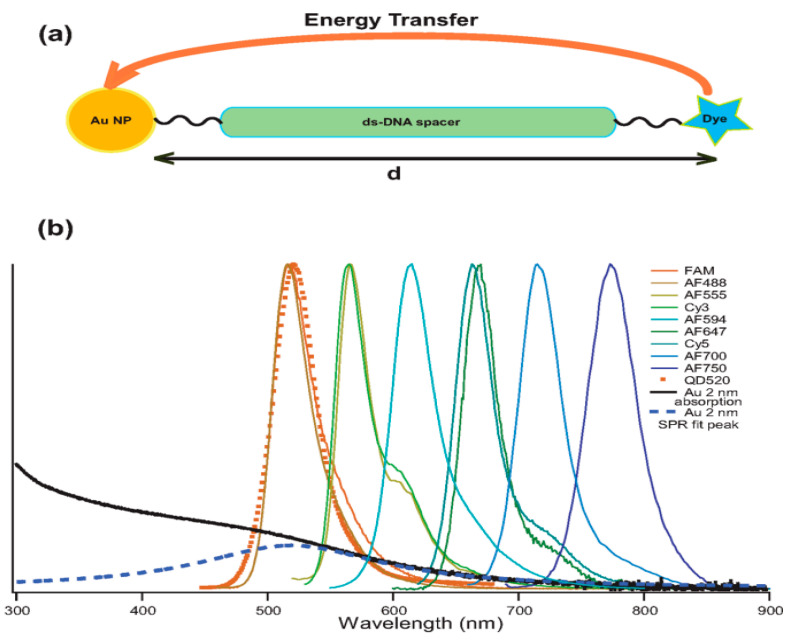
(**a**) Energy transfer schematic drawing. (**b**) Normalized donors’ PL spectra and the extinction of AuNPs [92].

**Figure 22 materials-16-03083-f022:**
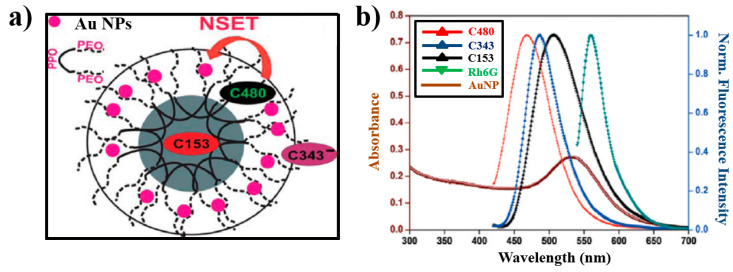
(**a**) Schematic illustration of P123 micelles and probe molecules at different positions. (**b**) Typical normalized spectra of P123 micelles and gold NPs [93].

**Figure 23 materials-16-03083-f023:**
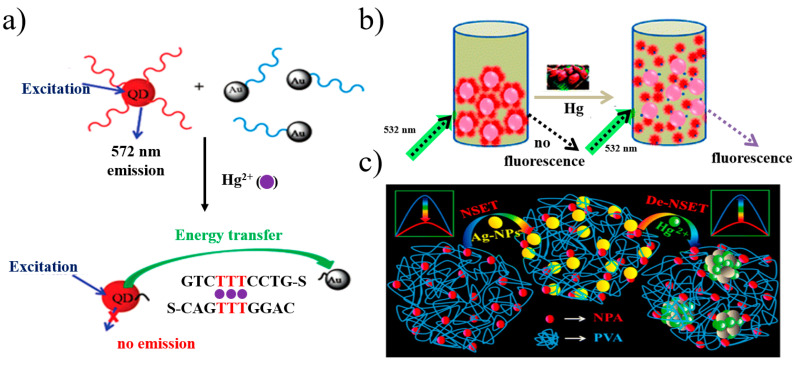
(**a**) Schematic diagram of the fluorescent sensor for Hg^2+^ detection [95]. (**b**) Schematic diagram of Hg^2+^ ion detection through NSET between rhodamine B and gold NPs [96]. (**c**) Schematic diagram of NSET modified by AuNP-doped PVA [97].

**Figure 24 materials-16-03083-f024:**
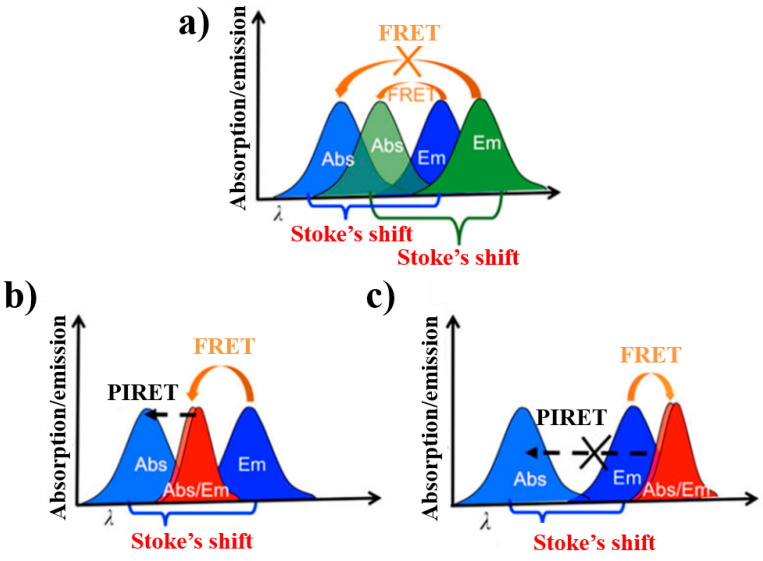
Schematic diagrams of energy transfer under different conditions in the NP–molecule hybrid state [99].

**Figure 25 materials-16-03083-f025:**
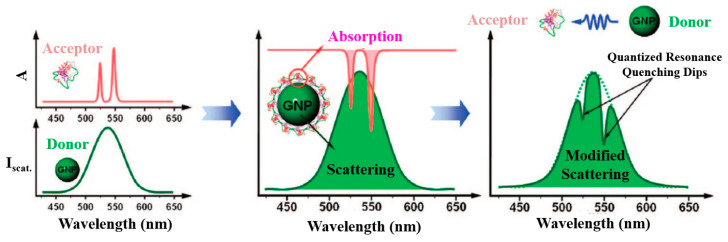
PIRET-based molecular imaging systems [102].

**Figure 26 materials-16-03083-f026:**
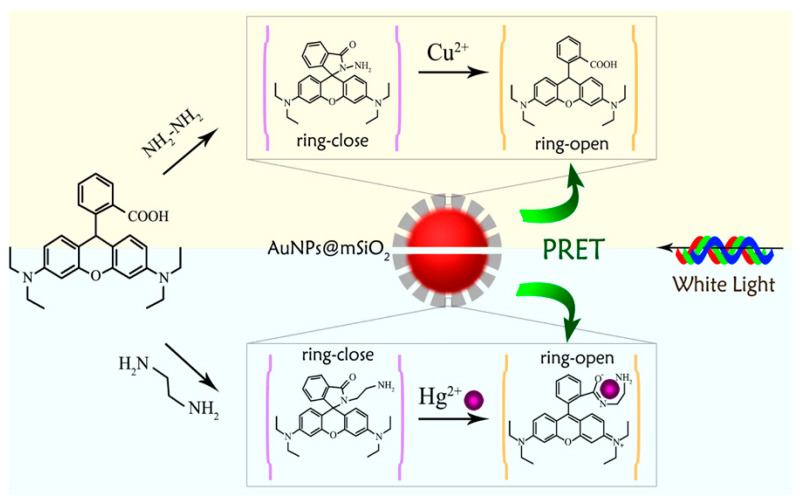
Illustration of the core–shell structure of AuNPs coated with mesoporous silica and the injection of rhodamine derivatives [103].

**Figure 27 materials-16-03083-f027:**
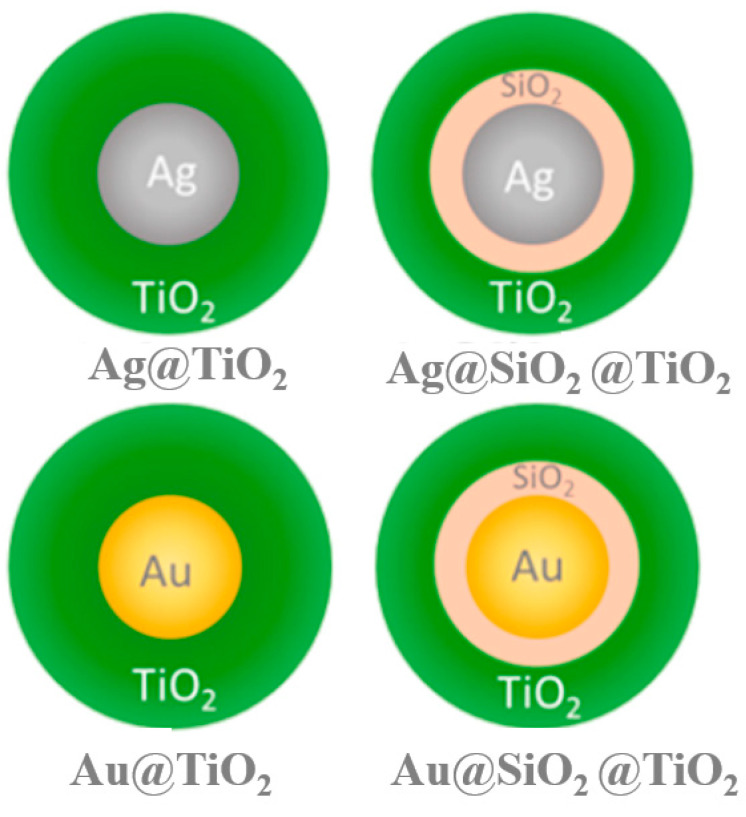
Schematic drawing of metal @ TiO_2_ and Ag @ SiO_2_ @ TiO_2_ structure [104].

**Figure 28 materials-16-03083-f028:**
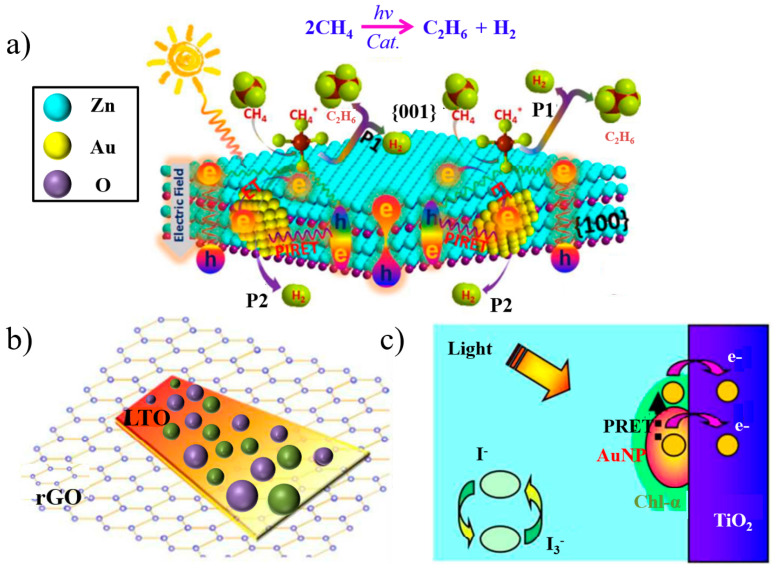
(**a**) Schematic description of idealized photocatalysts. H* is reduced by hot electrons to H_2_ (P1 process), or released as H_2_ on Au NPs via a hydrogen spillover pathway (P2 process) [108]. (**b**) Image of the Au@Pt-NLTO/rGO hybrid nanostructure. Adapted from Ref. [109]. (**c**) Charge and energy transfer channel(s) from hybrid plexciton to cell anodes [110].

**Figure 29 materials-16-03083-f029:**
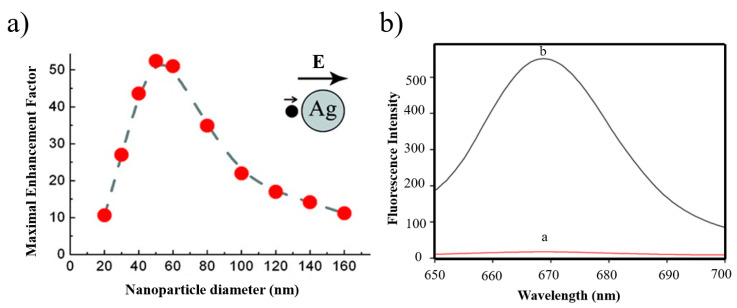
(**a**) The relationship between the fluorescence enhancement factor and the NPs’ diameter. The emission wavelength is 530 nm [119]. (**b**) Typical fluorescence spectra of a nanoflare (curve a) and control sample (curve b) [120].

**Figure 30 materials-16-03083-f030:**
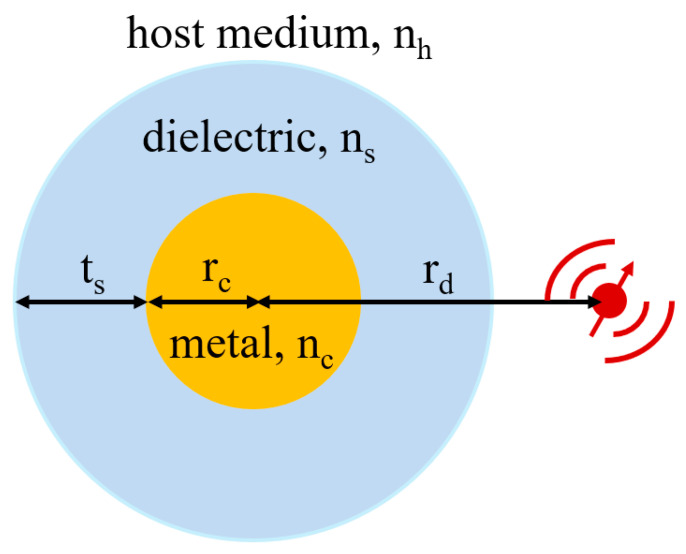
Emission process with dipole radiation. Adapted from Ref. [121].

**Figure 31 materials-16-03083-f031:**
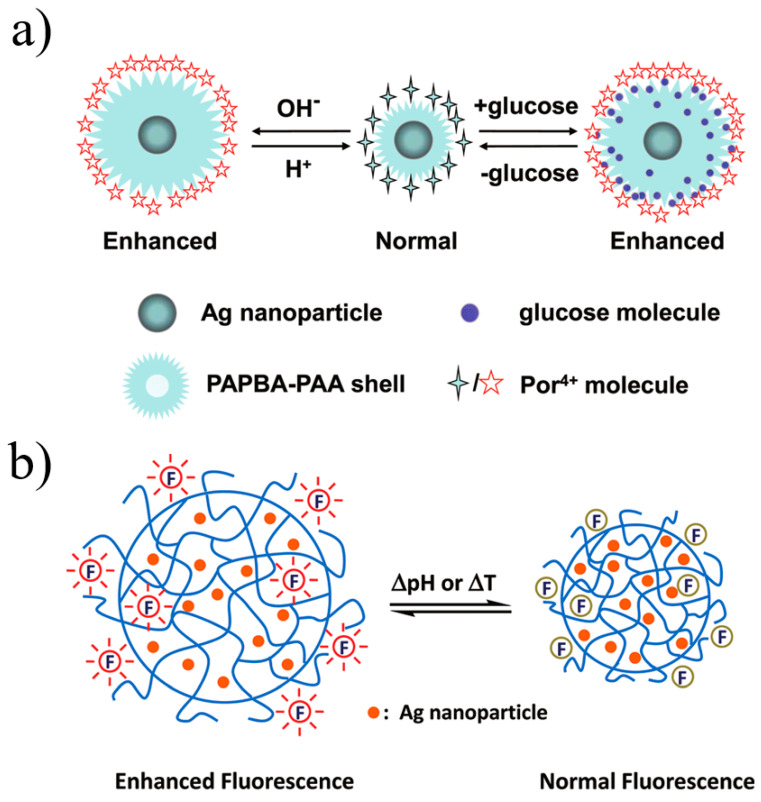
(**a**) Schematic illustration of the pH- and glucose-sensitive hybrid system [122]. (**b**) Schematic illustration of silver core hybrid PNIPAM-co-PAA microgels [123].

**Figure 32 materials-16-03083-f032:**
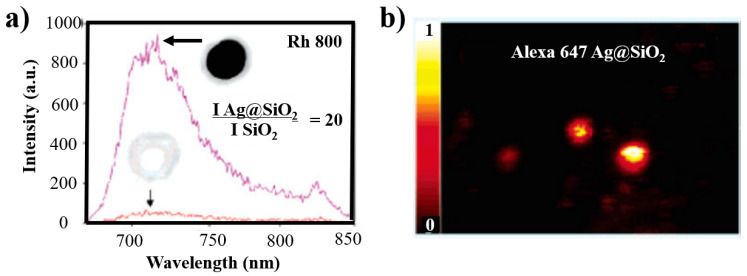
(**a**) Fluorescence emission of Rh800-doped Ag@SiO_2_. (**b**) Scanning confocal images (20 µm × 20 µm) of Alexa 647 Ag@SiO_2_ [124].

**Figure 33 materials-16-03083-f033:**
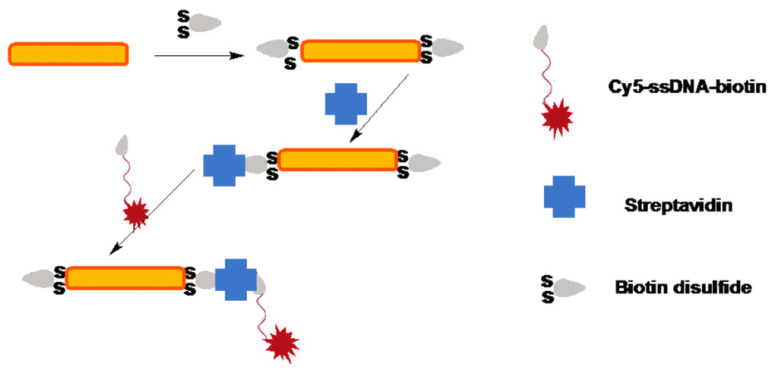
Combination of gold nanorods with CTAB and Cy5 ssDNA [127].

**Figure 34 materials-16-03083-f034:**
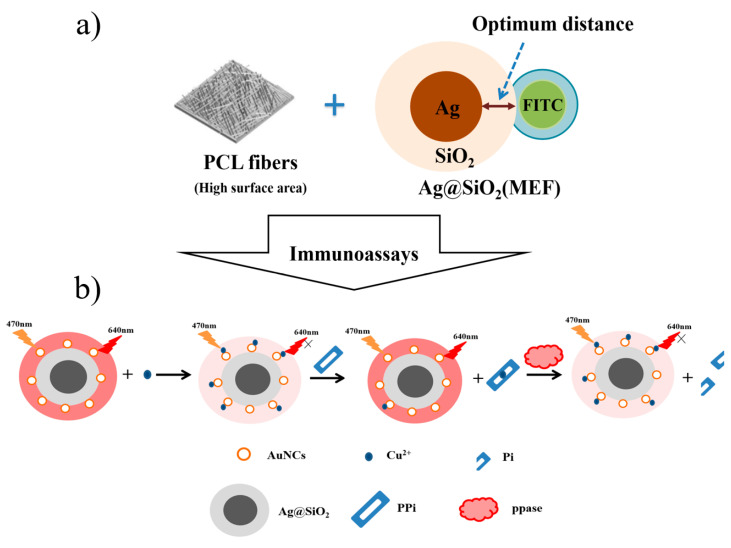
(**a**) Schematic illustration of MEF-based Ag@SiO_2_ platform. Adapted from Ref. [129]. (**b**) Schematic drawing of AuNCs conjugated to the surface of the Ag@SiO_2_ core–shell hybrid nano-system. Adapted from Ref. [130].

**Figure 35 materials-16-03083-f035:**
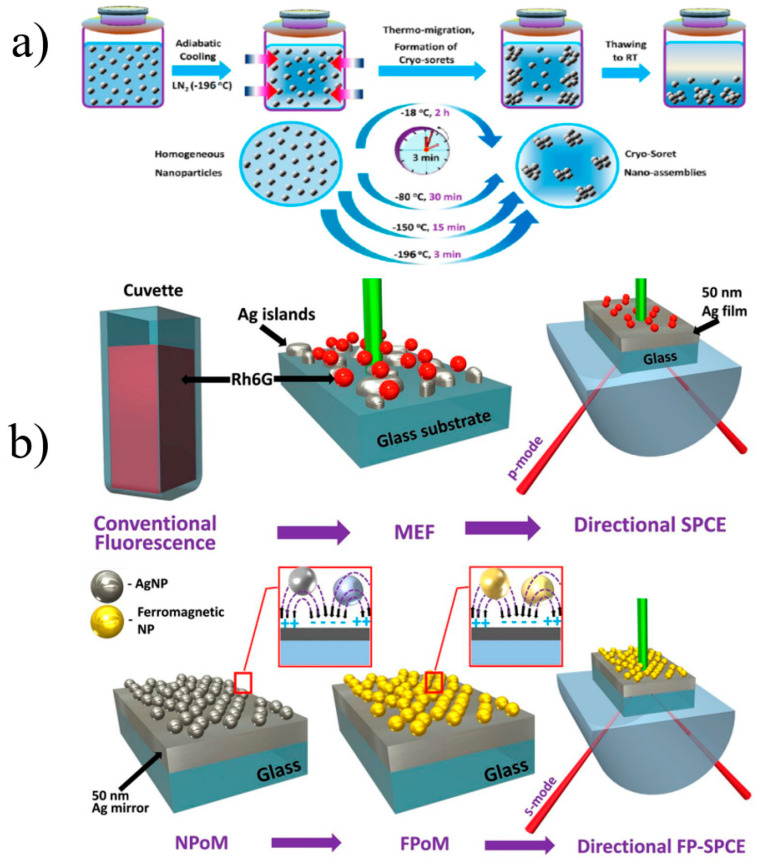
(**a**) A conceptual schematic of the Soret nanoassembly synthesis [141]. (**b**) Progression from conventional fluorescence to MEF and to directional SPCE and progression from nanoparticle-on-mirror (NPoM) to FPoM and to directional ferroplasmon-coupled SPCE [143].

**Figure 36 materials-16-03083-f036:**
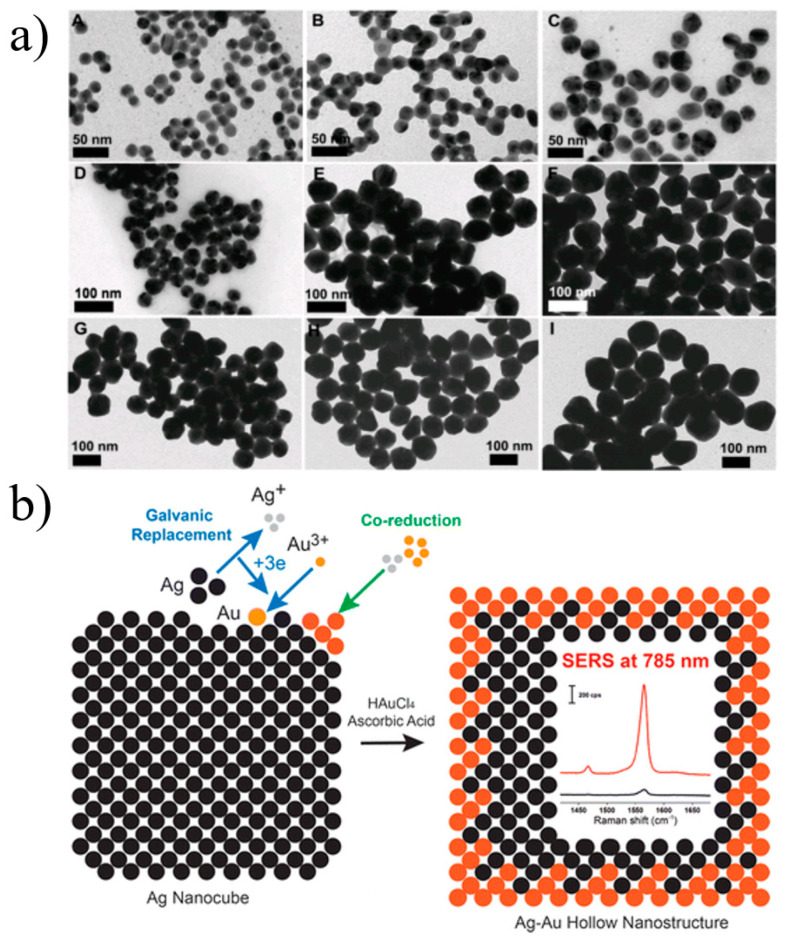
(**a**) TEM images of Au @ Ag core–shell nanospheres from Au NPs of different sizes. [(**A**–**C**) 15 nm, (**D**–**F**) 32 nm, and (**G**–**I**) 55nm after 3 (**A**, **D**, and **G**), 5 (**B**, **E**, and **H**), and 10 (**C**, **F**, and **I**) additions of AgNO_3_] [152]. (**b**) Schematic illustration of Ag-Au hollow nanostructures [153].

**Figure 37 materials-16-03083-f037:**
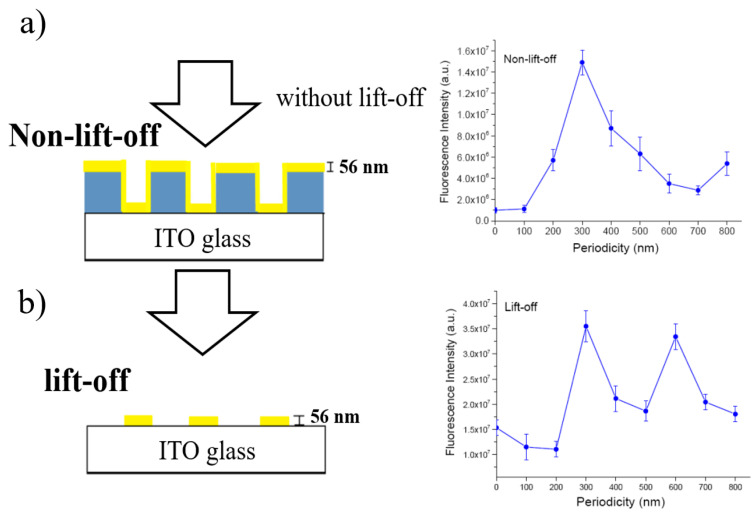
Schematic diagram of SERS and fluorescence signals of two nanograting structures and the periodicity dependence of fluorescence intensity on the two structures [154]. (**a**) A layer of 56 nm gold thin film is subsequently sputtered on the nanostructure (left); Periodicity dependence on nonlift-off nanogratings (right). (**b**) The photoresist is removed by a lift-off process, which is discrete gold nanostripes (left); Periodicity dependence on lift-off nanogratings (right).

**Figure 38 materials-16-03083-f038:**
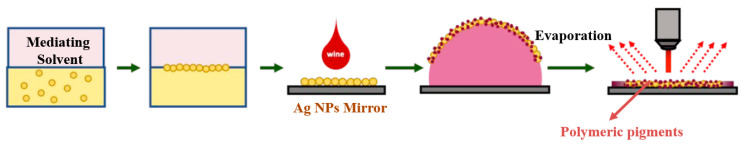
Schematic illustration of the chemical information of red wines with SERS through several sample preparation methods [155].

**Figure 39 materials-16-03083-f039:**
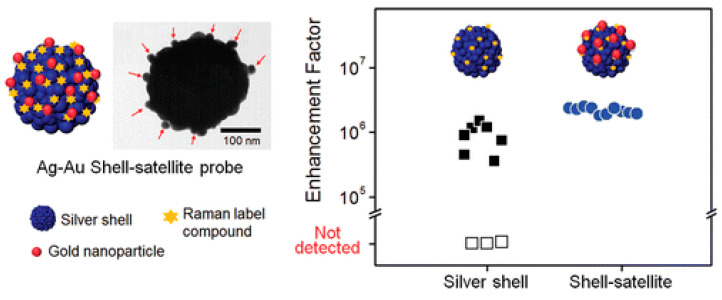
Drawing of Ag-Au heterosatellitallic shell–satellite structure and its optical properties [156].

**Figure 40 materials-16-03083-f040:**
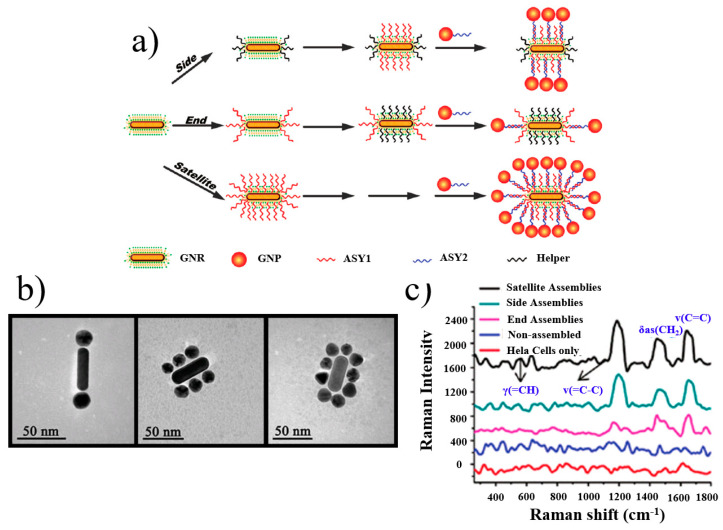
(**a**) Schematics of three types of NP assemblies. (**b**) Typical TEM images. (**c**) Raman spectra of three types of assemblies [157].

**Figure 41 materials-16-03083-f041:**
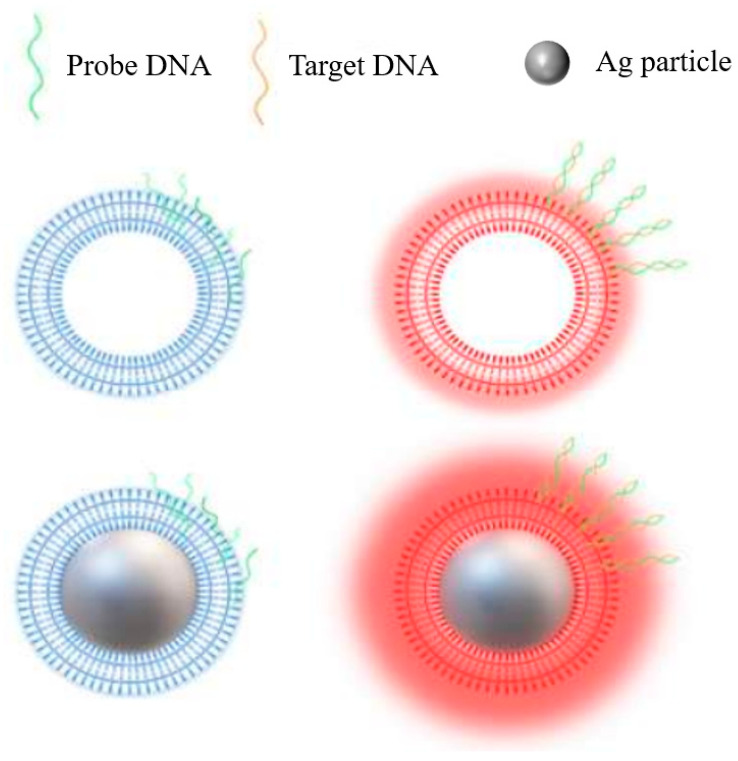
Schematic drawing of the DNA hybridization process [158].

**Figure 42 materials-16-03083-f042:**
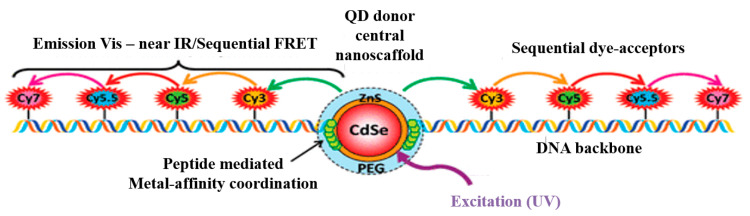
Schematics of DNA/peptide sequences and peptide–DNA chemoselective ligation [164].

**Figure 43 materials-16-03083-f043:**
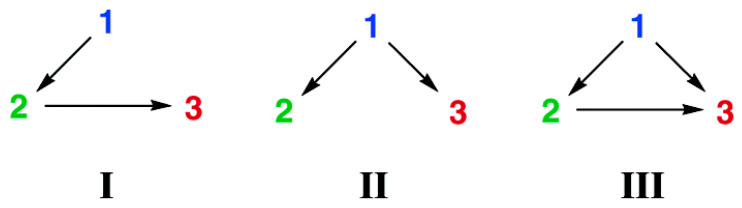
Schematics of three energy transfer schemes [165].

**Figure 44 materials-16-03083-f044:**
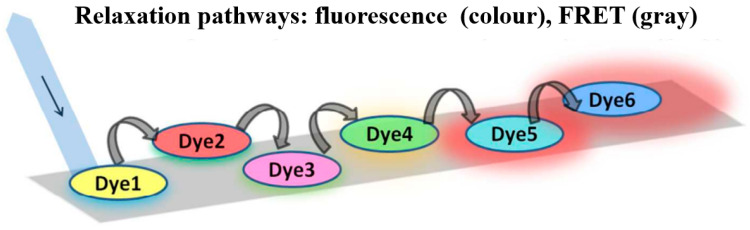
The possible relaxation process of the six dyes [166].

**Figure 45 materials-16-03083-f045:**
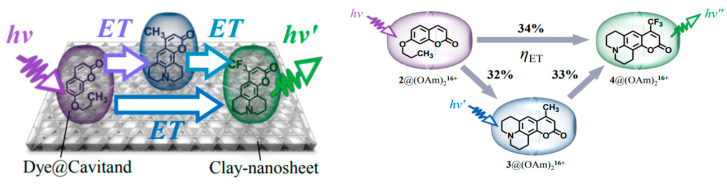
Multistep energy transfer reaction system [167].

**Figure 46 materials-16-03083-f046:**
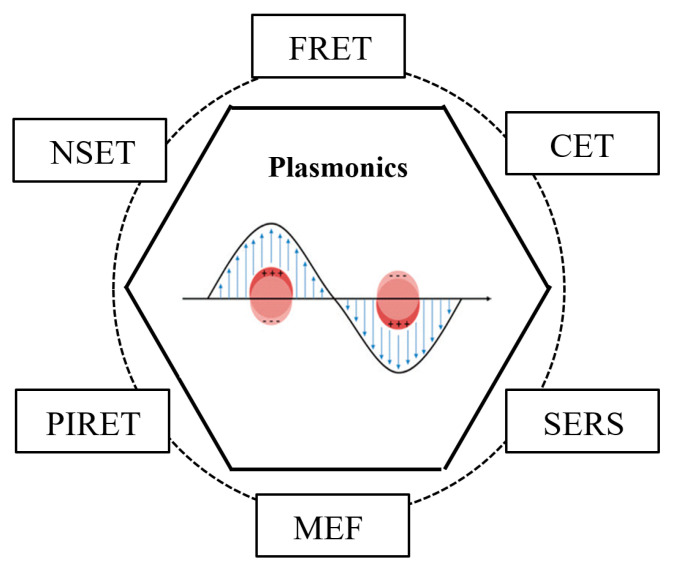
The main resonance energy transfer process involving plasmonic noble NPs.

**Table 1 materials-16-03083-t001:** Different resonance energy transfer processes involving plasmonic noble metal NPs with advantages, disadvantages and applications.

S1. No.	RET Process	Principles	Advantages	Disadvantages	Applications	Ref.
1	FRET	Due to the overlap of spectra, efficiency is inversely proportional to the sixth power of the distance.	FRET shows remarkable features, such as simple operation, fast detection and sensitivity, and is an important phenomenon for detecting donor–acceptor short-range-dependent interactions.	FRET depends on the probe’s interaction with the environment and the uncertainty of the position and orientation relative to the biomolecules, and the active distance is not more than 10 nm.	Biological analysis, sensing and imaging.	[50,51,57]
2	NSET	Efficiency is inversely proportional to the fourth power of the distance, which is suitable for long-range transfer.	NSET overcomes the distance limitation and shows unique sensitivity when the distance is more than 10 nm; moreover, it reveals good selectivity for representative metal ions.	NSET correlates with the size of metallic particles; small changes in size show apparent effects.	Optical biological imaging, metal ion detection and energy storage.	[79,85]
3	PIRET	Energy transfers from the plasma to the adjacent semiconductor through dipole–dipole interactions.	Large absorption coefficient, wide absorption spectrum and good stability.	The PIRET process depends on the overlap between the scattering/emission of NPs and the absorption spectrum of fluorophores.	Spectroscopy, biosensors and energy storage devices.	[99]
4	MEF	It is regarded as a mirrored dipole, which is due to the resonance between metallic NPs’ surface plasmons and fluorescent molecules.	High sensitivity, high efficiency and convenience, strong electric field enhancement and low loss characteristic of the dielectric in the structure.	High production cost.	Nanophotonics, plasmonic sensing and PH-sensitive devices.	[115,116,118]
5	SERS	The Raman scattering signal is greatly enhanced by molecules adsorbed on the metal NPs’ surfaces; further, it shows excellent plasmonic efficiency in the visible-light range.	SERS provides a larger surface area for sites of electromagnetic enhancement; with good biocompatibility and easy modification, it has significant advantages.	Lack of reliability and universality.	In life science, cell monitoring and optical physics.	[148,150]
6	CET	A higher efficiency in a wide wavelength range, a larger Stokes shift, the easier detection of the final acceptor, etc.	The roles of the energy donor and acceptor are no longer fixed in the system, with a higher efficiency in a wide wavelength range, a larger Stokes shift and the easier detection of the final acceptor.	It may need a longer reaction time.	Bionic optical synthesis, bioactivity analysis and cell membrane multivalence.	[160,161]

## Data Availability

Not applicable.

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
