# Peer review of "Principles and Applications of Resonance Energy Transfer Involving Noble Metallic Nanoparticles"

_materials, 2023, doi:10.3390/ma16083083_

Round 1

Reviewer 1 Report

This manuscript reports on a review of principle and applications of the resonance energy transfer involving noble metallic nanoparticles. My comments are as follow:

1.     The abstract section is too short. In the abstract section, suggest authors to include the problem statement of why this review is critical to be carried out, what is the main contribution of this paper, and highlight all the importance information in regard to this review paper.

2.     Suggest including summary in every section to highlight the main point or content from every section. 

3.     Suggest replacing Figs. 1, 4, 7, 10, 11, 12, 19, 24, 25, 27, 29, 31, 32, 34, 35, 36, 37, 39, 40, 43, 45, 46, 47, 49, 50, 51, 52, 53, 54, 56, 57 with better resolution figures.

4.  There are a few techniques under section 3 such as FRET, NSET, PIRET, and so forth. Authors make comparison of performance for these techniques. Suggest summarizing all the performance comparison for each technique in a table to highlight the main point.

I would also like to suggest the authors to reorganized the content and highlight the main contribution of the work.

Author Response

Response to Reviewer-1

Comment 1): The abstract section is too short. In the abstract section, suggest authors to include the problem statement of why this review is critical to be carried out, what is the main contribution of this paper, and highlight all the importance information in regard to this review paper.

Author reply: We appreciate the reviewer for her/his positive comments, we have revised our abstract section, which is marked in red in Line 12-25.

Comment 2):  Suggest including summary in every section to highlight the main point or content from every section. 

Author reply: We thank the reviewer for this comment. We have added summary in every section to highlight the main point from every section, which is marked in red in Line 81-85 and Line 129-133.

Comment 3):  Suggest replacing Figs. 1, 4, 7, 10, 11, 12, 19, 24, 25, 27, 29, 31, 32, 34, 35, 36, 37, 39, 40, 43, 45, 46, 47, 49, 50, 51, 52, 53, 54, 56, 57 with better resolution figures.

Author reply: Thanks for the reviewer’s good comment to this manuscript. We have replaced these Figures (NO. 6 (origin No. 7), 9 (origin No. 10), 10 (origin No. 11), 19 (b) (origin No. 24), 30 (origin No. 40), 34 (a) (origin No. 45) and 34 (b) (origin No. 46) are all redrawn) with better resolution in the revised manuscript.

Comment 4):  There are a few techniques under section 3 such as FRET, NSET, PIRET, and so forth. Authors make comparison of performance for these techniques. Suggest summarizing all the performance comparison for each technique in a table to highlight the main point.

Author reply: We thank the reviewer for the comment and suggestion. We have added the summarizing all the performance comparison for each technique in Table 1 in the revised manuscript, which is marked in red in Line 898.

Comment 4): I would also like to suggest the authors to reorganized the content and highlight the main contribution of the work.

Author reply: Thanks for the reviewer’s good comment to this manuscript. We have reorganized the content and highlight the main contribution of the work in in the revised manuscript.

Reviewer 2 Report

The article titled ‘Principle and Applications of the Resonance Energy Transfer 2 Involving Noble Metallic Nanoparticles’ by Tian and co-workers (He et al.) is an interesting review that is of immense utility to the broad audience of nanoscience, luminescence, photo-plasmonics, and biosensing technologies. To begin with the authors, provide a brief discussion on broad arena of plasmonics. Fruther, the article discusses the major domains of FRET, NSET, PIRET, MEF, SERS, CET and is succinctly and wonderfully captured in Figure 58. The article is well-planned, organized and well-written with several examples from the literature. The contents are intriguing and hence suitable for consideration for revision.

The total number of figures are redundantly exaggerated to >50! However, a review paper is not simply collection of each figure from different article and combining them into an article. The figures with single aspect from different papers should be combined into one figure. Also, authors should present insightful feedback from these earlier articles, without just presenting them to the journal audience. The futuristic scope, advantages and disadvantages and authors critical evaluation comprises a successful review, as these aspects are missing. This interesting work may be considered for publication, provided the authors address the below mentioned major/serious comments.

1.     Please rewrite confusing sentences ‘Due to the optical properties of noble metals, especially the optical scale and other characteristics in the process of energy transfer, possible future applications of micro lasers, quantum information storage devices and micro nano processing’; ‘The optical properties like extinction and scattering of metallic particles accounting for the surface plasmon resonance (SPR) were theoretically by the groundbreaking work of Mie in 1908’ and many more such sentences should be carefully revised.

2.     Resolution of certain figures are very low, and the contents cannot be read. Please improve them.

3.     Introduction section is poor. The motivation behind presenting this review is not presented. Important and relevant articles should be mentioned: ACS Appl. Mater. Interfaces 2020, 12, 30, 34323–34336; Micromachines 2021, 12(5), 492; Materials 2022, 15(23), 8677.

4.     There are several reviews in the similar domain. How is this review different from the existing ones needs to be clearly presented.

5.     What is the difference between ‘Förster resonance energy transfer (FRET)’ and ‘fluorescence resonance energy transfer’.

6.     The authors state ‘which makes the material different from bulk materials in optics and thermology, are as follows:’. However, the reviewer does not see any discussion on the thermology aspect. Please discuss what is claimed. However, it is understandable that thermology details are beyond the scope of the work and hence the claim may be removed (if not discussed).

7.     The futuristic scope and outlook are missing. The enhanced fluorescence signal can be used for smartphone-based surface plasmon-coupled emission (SPCE) for early diagnosis and these details should be mentioned.

8.     The authors mentioned ‘metallic NPs are widely used as inorganic fluorescent materials in FRET system’. It should be clearly stated as to this is true under what conditions. How plasmonic NPs can function as fluorescent materials? Do they have intrinsic fluorescence? The reviewer believes that the authors intend to state that the ‘metallic NPs are widely used as substrates and studied with inorganic fluorescent materials to generate a FRET system’. Such technically misleading sentences should be removed or modified across the manuscript.

9.     Several places in the manuscripts, the subscripts and superscripts are not clearly formatted. This is very confusing to the readers [for instance, SiO2, 3×10-8 M; Pb2+]. Please rectify the same.

10.  There are several formatting errors that needs to be revised ‘with different gold and QDs concentration. [63].’ Please provide a space gap between the SI unit and the value ‘10nm’.

11.  It is advisable to combine the figures under the same domain into a single figure, instead of discussing each figure from different paper separately. For instance, figures 18,19 and 20 can be combined to give a summary figure of the same topic of discussion. Similarly, several other figures should be combined.

12.  The labelling in figure 46 is not correct. Please rectify, Cu2+ and AuNCs. Similarly, all the other figures should be revised accordingly.

13.  The section on MEF is not comprehensive as compared to other sections. Please add discussion from recent works on cryosorets, Nd2O3-Ag, Nd2O3-gold soret, ferroplasmon-on-mirror and related platforms for MEF, with applications for biosensing technologies.

14.  What is the meaning of ‘plasma’ noble NPs? It should be plasmonic NPs from noble metals.

15.  This is an interesting review paper. However, the presentation is very poor. The authors present several figures, however from different papers as individual figures. Figures from the same underlying concept should be combined and presented succinctly. The final table should present the advantages and disadvantages of the processes. References are missing in the table.

Author Response

Response to Reviewer-2

The article titled ‘Principle and Applications of the Resonance Energy Transfer 2 Involving Noble Metallic Nanoparticles’ by Tian and co-workers (He et al.) is an interesting review that is of immense utility to the broad audience of nanoscience, luminescence, photo-plasmonics, and biosensing technologies. To begin with the authors, provide a brief discussion on broad arena of plasmonics. Further, the article discusses the major domains of FRET, NSET, PIRET, MEF, SERS, CET and is succinctly and wonderfully captured in Figure 58. The article is well-planned, organized and well-written with several examples from the literature. The contents are intriguing and hence suitable for consideration for revision.

The total number of figures are redundantly exaggerated to >50! However, a review paper is not simply collection of each figure from different article and combining them into an article. The figures with single aspect from different papers should be combined into one figure. Also, authors should present insightful feedback from these earlier articles, without just presenting them to the journal audience. The futuristic scope, advantages and disadvantages and authors critical evaluation comprises a successful review, as these aspects are missing. This interesting work may be considered for publication, provided the authors address the below mentioned major/serious comments.

Author reply: We appreciate the reviewer for her/his positive comments, a lucid summary of our novelty, and potential recommendations. We have addressed the questions point-by-point raised by the reviewer as below.

Comment 1): Please rewrite confusing sentences ‘Due to the optical properties of noble metals, especially the optical scale and other characteristics in the process of energy transfer, possible future applications of micro lasers, quantum information storage devices and micro nano processing’; ‘The optical properties like extinction and scattering of metallic particles accounting for the surface plasmon resonance (SPR) were theoretically by the groundbreaking work of Mie in 1908’ and many more such sentences should be carefully revised.

Author reply: We thanks for the reviewer’s good comment to this manuscript. We have revised these confusing sentences in our manuscript carefully, which is marked in red in Line 14-18, Line 93-95.

Comment 2): Resolution of certain figures are very low, and the contents cannot be read. Please improve them.

Author reply: Thanks for the reviewer’s good comment to this manuscript. We have replaced these Figures with low resolution, and Figures (NO. 6 (origin No. 7), 9 (origin No. 10), 10 (origin No. 11), 19 (b) (origin No. 24), 30 (origin No. 40), 34 (a) (origin No. 45) and 34 (b) (origin No. 46) are all redrawn with better resolution in the revised manuscript.

Comment 3): Introduction section is poor. Important and relevant articles should be mentioned: ACS Appl. Mater. Interfaces 2020, 12, 30, 34323–34336; Micromachines 2021, 12(5), 492; Materials 2022, 15(23), 8677.

Author reply: We thank the reviewer for the comment and suggestion. We have added the important and relevant articles in the revised manuscript, with the number are 35, 31 and 32.

Comment 4): There are several reviews in the similar domain. How is this review different from the existing ones needs to be clearly presented.

Author reply: Thanks for your suggestion. We have reviewed lots of references about the resonance energy transfer, but none has been as comprehensive as our manuscript. Gao et al. [TrAC Trends in Analytical Chemistry, 2020, 124, 115805] summarized the analytical applications of just both the NSET and PIRET technologies, there was little relevant content about FRET, MEF, SERS and CET. Malekzad H. et al. [Nanotechnology Reviews, 2017, 6(3), 301] and Yeo et al. [Journal of Biomedical Nanotechnology, 2014, 10, 2722] focused on the different classes of biosensors. Further, Hang et al. [Chem. Soc. Rev.,2022,51,329] dealt with only surface-enhanced Raman scattering (SERS) in in vitro point-of-care testing and in vivo bio-imaging.

Comment 5): What is the difference between ‘Förster resonance energy transfer (FRET)’ and ‘fluorescence resonance energy transfer’.

Author reply: Thanks for your suggestion. Actually, Förster resonance energy transfer sometimes is also referred as fluorescence resonance energy transfer [Journal of photochemistry and photobiology C: Photochemistry Reviews, 2011, 12(1), 20; Analytical Methods, 2020, 46, 5532], and we have cited in the revised manuscript, which is marked in red in Line 140.

Comment 6): The authors state ‘which makes the material different from bulk materials in optics and thermology, are as follows:’. However, the reviewer does not see any discussion on the thermology aspect. Please discuss what is claimed. However, it is understandable that thermology details are beyond the scope of the work and hence the claim may be removed (if not discussed).

Author reply: Thanks for the reviewer’s good comment to this manuscript. We have removed the thermology details for which are beyond the scope of this work.

Comment 7): The futuristic scope and outlook are missing. The enhanced fluorescence signal can be used for smartphone-based surface plasmon-coupled emission (SPCE) for early diagnosis and these details should be mentioned.

Author reply: We thank the reviewer for the comment and suggestion. We have added the details about enhanced fluorescence signal, which can be used for smartphone-based surface plasmon-coupled emission (SPCE) in the revised manuscript, which is marked in red in Line 730-751.

Comment 8): The authors mentioned ‘metallic NPs are widely used as inorganic fluorescent materials in FRET system’. It should be clearly stated as to this is true under what conditions. How plasmonic NPs can function as fluorescent materials? Do they have intrinsic fluorescence? The reviewer believes that the authors intend to state that the ‘metallic NPs are widely used as substrates and studied with inorganic fluorescent materials to generate a FRET system’. Such technically misleading sentences should be removed or modified across the manuscript.

Author reply: Thanks for the reviewer’s good comment to this manuscript. We have modified this sentence in the revised manuscript, which is marked in red in Line 175-176.

Comment 9): Several places in the manuscripts, the subscripts and superscripts are not clearly formatted. This is very confusing to the readers [for instance, SiO2, 3×10-8 M; Pb2+]. Please rectify the same.

Author reply: We thank the reviewer for the comment and suggestion. We have clearly formatted the subscripts in the revised manuscript, which is marked in red in Line 174, 236-239, 246, 271, 281, 323-325, 379, 477-485, 548, 571-572, 581-589, 608, 644-648, 658, 678-679, 685, 713-723,794, and 816.

Comment 10): There are several formatting errors that needs to be revised ‘with different gold and QDs concentration. [63].’ Please provide a space gap between the SI unit and the value ‘10nm’.

Author reply: We thank the reviewer for the comment and suggestion. We have added a space gap between the SI unit and the value in the revised manuscript.

Comment 11): It is advisable to combine the figures under the same domain into a single figure, instead of discussing each figure from different paper separately. For instance, figures 18,19 and 20 can be combined to give a summary figure of the same topic of discussion. Similarly, several other figures should be combined.

Author reply: Thanks for the reviewer’s good comment to this manuscript. We have combined the figures which are the same topic of discussion on the revised manuscript.

Comment 12): The labelling in figure 46 is not correct. Please rectify, Cu2+ and AuNCs. Similarly, all the other figures should be revised accordingly.

Author reply: We thank the reviewer for the comment and suggestion. We have redrawn the figure 46 for the unrequired permission, and we revised the labelling in the revised manuscript, which is marked in red in Line 726.

Comment 13):  The section on MEF is not comprehensive as compared to other sections. Please add discussion from recent works on cryosorets, Nd2O3-Ag, Nd2O3-gold soret, ferro plasmon-on-mirror and related platforms for MEF, with applications for biosensing technologies.

Author reply: Thanks for the reviewer’s good comment to this manuscript. We have added the related content in the revised manuscript, which is marked in red in Line 730-751.

Comment 14): What is the meaning of ‘plasma’ noble NPs? It should be plasmonic NPs from noble metals.

Author reply: We thank the reviewer for the comment and suggestion. We have replaced “plasma noble NPs” to “plasmonic noble NPs” in the revised manuscript, which is marked in red in Line 895-898.

Comment 15): This is an interesting review paper. However, the presentation is very poor. The authors present several figures, however from different papers as individual figures. Figures from the same underlying concept should be combined and presented succinctly. The final table should present the advantages and disadvantages of the processes. References are missing in the table.

Author reply: Thanks for the reviewer’s good comment to this manuscript. We have combined and presented the Figures from the same underlying concept, in addition, the advantage, disadvantages and References are all added in the revised manuscript, which is marked in red in Line 899.

Reviewer 3 Report

This paper reviewed the  resonance energy transfer involving noble metallic NPs which is a very crucial topic in nanoparticle applications for revealing the mechanism involved. Further, the authors also classified the phenomena into 5 types: FRET, PIRET, NSET, SERS and cascade FRET.  This classification is a great knowledge for the reader to understand firmly about RET. The content is highly valuable for a greater reader, but the manuscript is lack of explorations, insight and discussions about the topic despite of many works being referred. No further comments for the article content since it is mostly a citation from other works, but format, grammar and typo are still an issue.  Also many figures are cited from other works without courtesy from the publishers nor self-reconstruction; it may be considered as a plagiarism. Major improvements in terms of insight of the topic and article layout should be addressed before publication. 
Here are some lazy typo found in the article:

pp 9 line 276: Ahmad Aflzalina et al. [65] ... --> Afzalina et al.[65]

pp 367 - 371: ineffective sentences. It should be break into several sentences.

pp 13 line 380: What is R0? is it the same as R(subscripted 0), Foster distance? Please consistent with the typing of this parameter in the entire article. Also for others such as d0, Mg2+, Hg2+, Cu2+, SiO2, TiO2, Fe2O3, La2Ti2O7, HAuCl4..etc

pp 30 line 829 : Figure 49 was the SERS and fluorescence signals from two kinds.....In fact, Figure 49 is only a schematic of the two nanogratings without their respective fluorescence spectra.

pp 31 line 862 - 865 : Please check the sentences with a proper grammar and typo.There are lot of paragraphs consisted of one sentence only, e.g pp 34 line 924-927, pp 931 - 933, and many others. Those paragraphs need to be more elaborated. Or simply composing the two sentences into one larger paragraph.

Author Response

Response to Reviewer-3

This paper reviewed the resonance energy transfer involving noble metallic NPs which is a very crucial topic in nanoparticle applications for revealing the mechanism involved. Further, the authors also classified the phenomena into 5 types: FRET, PIRET, NSET, SERS and cascade FRET.  This classification is a great knowledge for the reader to understand firmly about RET. The content is highly valuable for a greater reader, but the manuscript is lack of explorations, insight and discussions about the topic despite of many works being referred. No further comments for the article content since it is mostly a citation from other works, but format, grammar and typo are still an issue. Also many figures are cited from other works without courtesy from the publishers nor self-reconstruction; it may be considered as a plagiarism. Major improvements in terms of insight of the topic and article layout should be addressed before publication. 

Author reply: We appreciate the reviewer for her/his positive comments, a lucid summary of our novelty, and potential recommendations. We have added explorations, insight and discussions about the topic, revised the format, grammar and typo, and combined and presented the Figures from the same underlying concept in the revised manuscript, and we addressed the questions point-by-point raised by the reviewer as below.

Comment 1):  pp 9 line 276: Ahmad Aflzalina et al. [65] ... --> Afzalina et al.[65]

Author reply: Thanks for the reviewer’s good comment to this manuscript. We have corrected “Ahmad Aflzalina et al. [65]” to “Afzalina et al.”, so as the same misrepresentation in our revised manuscript.

Comment 2):  pp 367 - 371: ineffective sentences. It should be break into several sentences.

Author reply: We thank the reviewer for the comment and suggestion. We have broken the ineffective sentences into several sentences in the revised manuscript, which is marked in red in Line 386-390.

Comment 3): pp 13 line 380: What is R0? is it the same as R(subscripted 0), Foster distance? Please consistent with the typing of this parameter in the entire article. Also for others such as d0, Mg2+, Hg2+, Cu2+, SiO2, TiO2, Fe2O3, La2Ti2O7, HAuCl4..etc

Author reply: Thanks for the reviewer’s good comment to this manuscript. We have added these typing of parameters in the entire articles in the revised manuscript, which is marked in red in Line 236, 271, 354, 376, 404, 412, 428, 477, 571,608 and 794.

Comment 4):  pp 30 line 829 : Figure 49 was the SERS and fluorescence signals from two kinds.....In fact, Figure 49 is only a schematic of the two nanogratings without their respective fluorescence spectra.

Author reply: We thank the reviewer for the comment and suggestion. We have added the respective fluorescence spectra in the revised manuscript, which is marked in red in Line 805-807.

Comment 5):  pp 31 line 862 - 865: Please check the sentences with a proper grammar and typo. There are lot of paragraphs consisted of one sentence only, e.g pp 34 line 924-927, pp 931 - 933, and many others. Those paragraphs need to be more elaborated. Or simply composing the two sentences into one larger paragraph.

Author reply: Thanks for the reviewer’s good comment to this manuscript. We have checked the sentences with a proper grammar in the revised manuscript, which is marked in red in Line 825-832, 884-888 and 889-890. Furthermore, we have improved the language in the revised manuscript.

Round 2

Reviewer 1 Report

Authors have amended the manuscript accordingly. I recommend this manuscript to be accepted in its present form.

Reviewer 2 Report

The authors have addressed the reviewer comments.

Reviewer 3 Report

The manuscript have a lot of improvement and the authors have followed all suggestions from the review. This manuscript can be published in PSF